# The mitophagy receptor NIX induces vIRF-1 oligomerization and interaction with GABARAPL1 for the promotion of HHV-8 reactivation-induced mitophagy

**Mai Tram Vo, Chang-Yong Choi, Young Bong Choi** *

Department of Oncology, Sidney Kimmel Comprehensive Cancer Center, Johns Hopkins University School of Medicine, Baltimore, Maryland, United States of America

* ychoi15@jhmi.edu

**Data Availability Statement:** The authors confirm that all data underlying the findings are fully available without restriction. The data supporting this study's findings are available in figshare at

## Abstract

Recently, viruses have been shown to regulate selective autophagy for productive infections. For instance, human herpesvirus 8 (HHV-8), also known as Kaposi's sarcoma-associated herpesvirus (KSHV), activates selective autophagy of mitochondria, termed mitophagy, thereby inhibiting antiviral innate immune responses during lytic infection in host cells. We previously demonstrated that HHV-8 viral interferon regulatory factor 1 (vIRF-1) plays a crucial role in lytic replication-activated mitophagy by interacting with cellular mitophagic proteins, including NIX and TUFM. However, the precise molecular mechanisms by which these interactions lead to mitophagy activation remain to be determined. Here, we show that vIRF-1 binds directly to mammalian autophagy-related gene 8 (ATG8) proteins, preferentially GABARAPL1 in infected cells, in an LC3-interacting region (LIR)-independent manner. Accordingly, we identified key residues in vIRF-1 and GABARAPL1 required for mutual interaction and demonstrated that the interaction is essential for mitophagy activation and HHV-8 productive replication. Interestingly, the mitophagy receptor NIX promotes vIRF-1-GABARAPL1 interaction, and NIX/vIRF-1-induced mitophagy is significantly inhibited in GABARAPL1-deficient cells. Moreover, a vIRF-1 variant defective in GABARAPL1 binding substantially loses the ability to induce vIRF-1/NIX-induced mitophagy. These results suggest that NIX supports vIRF-1 activity as a mitophagy mediator. In addition, we found that NIX promotes vIRF-1 aggregation and stabilizes aggregated vIRF-1. Together, these findings indicate that vIRF-1 plays a role as a viral mitophagy mediator that can be activated by a cellular mitophagy receptor.

## Author summary

In addition to their role in energy metabolism, mitochondria act as platforms for antiviral signaling that leads to apoptosis and type I interferon expression in response to virus infection. However, for their benefit, viruses have evolved strategies to attenuate mitochondria-mediated antiviral signaling, including selective autophagy of mitochondria

https://figshare.com/articles/figure/The_
mitophagy_receptor_NIX_induces_vIRF-1_
oligomerization_and_interaction_with_
GABARAPL1_for_the_promotion_of_HHV-8_
reactivation-induced_mitophagy/23488460.

**Funding:** This work was supported by National
Cancer Institute (NCI) grant R01CA214131 to Y.B.
C. and by National Institute of Allergy and
Infectious Diseases (NIAID) grant R21AI117168 to
Y.B.C. M.T.V. and C.Y.C received a salary from the
NCI grant. In addition, this research was partly
funded by a 2016-2017 developmental grant from
the Johns Hopkins University Center for AIDS
Research, an NIH-funded program (P30AI094189
to Y.B.C.), and an NIH Shared Instrumentation
grant (S10OD016374 to the JHU Microscope
Facility). The funders had no role in study design,
data collection and analysis, decision to publish, or
preparation of the manuscript.

**Competing interests:** The authors have declared
that no competing interests exist.

(termed mitophagy) that eliminates dysfunctional or superfluous mitochondria. We previously demonstrated that mitochondria-localized viral interferon regulatory factor 1 (vIRF-1) plays a role in the activation of mitophagy during human herpesvirus 8 (HHV-8) via interaction with the mitophagy proteins NIX and TUFM, thereby inhibiting antiviral responses and contributing to productive replication. However, it remains to elucidate the precise molecular mechanisms and regulation of vIRF-1-mediated mitophagy, particularly interactions with the core autophagy machinery. In this report, we discover that vIRF-1 interacts selectively with GABARAPL1 among ATG8 proteins in a non-canonical manner in lytically HHV-8-infected cells, and the mitophagy receptor NIX promotes vIRF-1 oligomerization and interaction with the ATG8 protein. Our results reveal the presence of a functional complex of vIRF-1/NIX/GABARAPL1 for the promotion of mitophagy and also provide the first evidence of the post-translational regulation of viral mitophagic protein by cellular mitophagy receptor.

## Introduction

Autophagy is a highly-conserved cellular degradation and recycling system crucial for cell survival during nutrient starvation [1]. It proceeds via the formation of double-membrane vesicles, termed autophagosomes, that engulf portions of the cytoplasm and deliver them to lysosomes. In addition to non-selective autophagy, various types of selective autophagy have evolved to remove specifically damaged or excess organelles or harmful protein aggregates, thus providing cellular quality control [2,3,4]. Selective autophagy is mediated by autophagy cargo receptors [5], such as p62/sequestosome 1 (SQSTM1), neighbor of BRCA1 gene (NBR1), optineurin (OPTN), nuclear dot protein 52 kDa (NDP52, also known as CALCOCO2), Tax1--binding protein 1 (TAX1BP1), BCL2/adenovirus E1B 19 kDa protein-interacting protein 3 (BNIP3), and NIP3-like protein X (NIX, also known as BNIP3L), which commonly contain a conserved motif termed the LC3-interacting region (LIR) [6] and bind directly to the mammalian autophagy-related gene 8 (ATG8) family of proteins, the microtubule-associated protein 1 light chain 3 A (MAP1LC3A or LC3A), LC3B, LC3C, the γ-amino-butyric acid receptor-associated protein (GABARAP), GABARAP-like 1 (GABARAPL1), and GABARAPL2, on the phagophore [7].

Selective autophagy is known to regulate or be regulated by virus infection. The host cells employ antiviral selective autophagy, termed viral xenophagy or virophagy, as a defense against invading viruses; components of the autophagy machinery recognize whole virion particles or virion components for clearance [8,9]. On the other hand, viruses have evolved protective mechanisms to evade antiviral autophagy or even to activate certain forms of selective autophagy for successful infection or replication [8,9,10]. Of note, selective autophagy of mitochondria, termed mitophagy, is highly activated in human herpesvirus 8 (HHV-8) productive infection and attenuates mitochondria-mediated antiviral responses, including apoptosis and innate immune responses [8].

HHV-8, also known as Kaposi's sarcoma-associated herpesvirus (KSHV), is an oncogenic virus associated with the development of Kaposi's sarcoma, primary effusion lymphoma, and multicentric Castleman's disease [11,12]. HHV-8 has two distinct infection stages, latency and lytic replication, in the host, which are causally associated with pathogenesis [13,14,15]. We previously demonstrated that HHV-8 activates mitophagy via its genome-encoded viral interferon regulatory factor 1 (vIRF-1), which promotes lytic replication. vIRF-1 can be localized in part to mitochondria by targeting detergent-resistant microdomains (DRM) in the outer

mitochondrial membrane (OMM) via its N-terminal region [16], where it interacts with the mitochondrial proteins NIX and the Tu translation elongation factor, mitochondrial (TUFM) and activates mitophagy [17,18]. The level of NIX protein is increased by more than twofold during HHV-8 lytic replication. Expression of both vIRF-1 and NIX, but not each gene alone, strongly induces mitochondrial clearance, and the LIR motif and dimerization of NIX are required for vIRF-1-activated mitophagy [17]. Moreover, we identified TUFM as a vIRF-1-interacting DRM protein and found that vIRF-1 promotes the dimerization of TUFM in the OMM, which interacts with the autophagy machinery protein conjugate ATG12-ATG5 [18]. The formation of autophagy-competent TUFM is associated with the inhibition of caspase-8-mediated apoptosis induced after lytic reactivation. Despite these significant findings, however, the precise molecular mechanisms underlying vIRF-1-activated mitophagy, including regulation of physical and functional connections with core components of the autophagic machinery, remain to be fully elucidated. Here, we report that vIRF-1 binds preferentially and directly to a group of ATG8 proteins *in vitro*, and its intracellular interaction with GABAR-APL1 is induced by virus lytic reactivation and NIX overexpression. Furthermore, we found that NIX is a crucial factor that promotes vIRF-1 oligomerization and associated mitophagy activation. Together, these findings highlight the novel role of vIRF-1 as a mitophagy mediator in HHV-8 lytic infection.

## Results

### vIRF-1 binds directly to human ATG8 orthologs

*In silico* inspection using the web-based tool iLIR [19] showed that vIRF-1 may have a potential LIR motif between the DNA-binding domain (DBD) and the IRF-association domain (IAD) (Fig 1A), raising the possibility that vIRF-1 can bind directly to ATG8 proteins. To test the prediction, we performed *in vitro* GST pull-down assays with purified recombinant proteins. The results showed that vIRF-1 could bind preferentially to LC3C, GABARAP, and GABARAPL1 but not to LC3A, LC3B, and GABARAPL2 (Fig 1B). As controls, vIRF-1 did not bind to GST alone, ubiquitin (Ub), and a small Ub-related modifier (SUMO) (Fig 1B). SUMO and ATG8 proteins are Ub-like proteins that share the Ub fold and the capacity to be conjugated to substrates through an enzymatic cascade [20].

We next examined *in cellulo* interactions of vIRF-1 with endogenous ATG8 proteins during lytic reactivation in TRExBCBL-1-RTA (simply termed iBCBL-1) cells, which are HHV-8-infected and doxycycline (Dox)-inducible for lytic switch protein RTA [21], using co-immunoprecipitation (co-IP) assays. ATG8 proteins are found mainly in the cytoplasm, while vIRF-1 is found in both the nucleus and the cytoplasm. To enhance the chance of detecting ATG8 proteins binding to vIRF-1, we used the cytoplasmic fraction of reactivated iBCBL-1 cells for vIRF-1-IP. The nuclear fraction was used as a control. The results showed that only GABAR-APL1 and, to a lesser extent, GABARAP were detected in the vIRF-1-IP complex precipitated from the cytoplasm (Fig 1C). Interestingly, the expression levels of GABARAPL and GABAR-APL1 steadily increased after lytic reactivation in iBCBL-1 cells (S1 Fig), suggesting that the GABARAP subfamily proteins may play an essential role in HHV-8 lytic infection. However, LC3C was not detected in the vIRF-1-IP complex (Fig 1C). This result may be due to a relatively low level of LC3C in reactivated iBCBL-1 cells (S1 Fig). To demonstrate the co-localization of vIRF-1 and GABARAPL1 in mitochondria, we performed immunofluorescence assays (IFA). To facilitate the detection of mitochondria-bound vIRF-1, reactivated iBCBL-1 cells were permeabilized with the mild detergent saponin before fixation to diffuse out free vIRF-1 from the cytoplasm as we performed previously [18]. The results showed that vIRF-1 co-localized with GABARAPL1 in the mitochondria of reactivated iBCBL-1 cells (Fig 1D). Together,

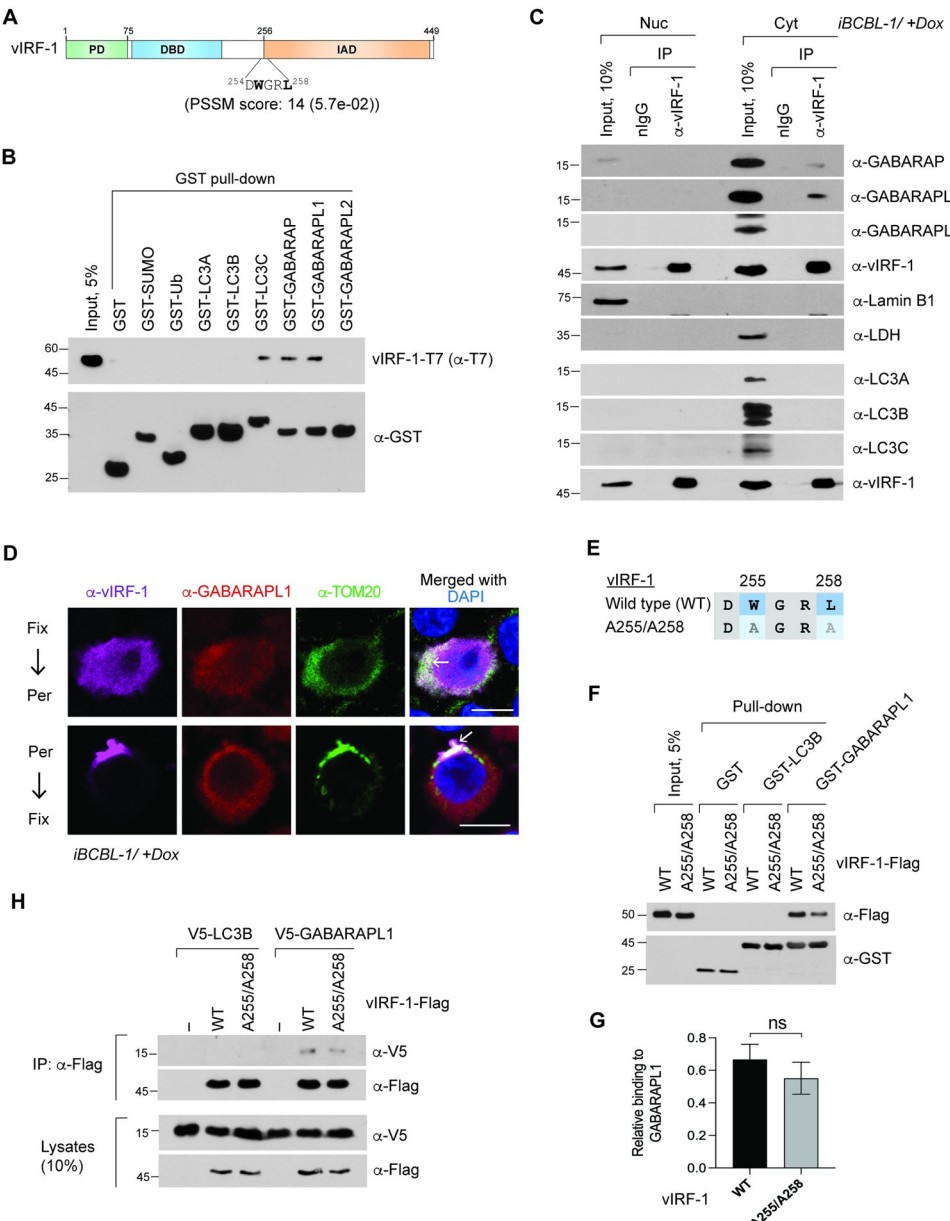

**Fig 1. vIRF-1 binds directly to GABARAPL1 in an LIR-independent manner. (A)** Schematic structure of vIRF-1: PD, Proline-rich domain; DBD, DNA-binding domain; and IAD, IRF-association domain. LC3-interacting regions (LIRs) were searched for using a position-specific scoring matrix (PSSM) via the web-based software iLIR, and a conserved LIR of vIRF-1 with the highest PSSM score was predicted at the N-terminal junction of the IAD, as noted. **(B)** GST pull-down assay to assess *in vitro* binding between vIRF-1 and ATG8 proteins (LC3A, LC3B, LC3C, GABARAP, GABARAPL1, and GABARAPL2). Purified recombinant T7-tagged vIRF-1 (vIRF-1-T7) was pulled down with 1μg of GST-fused ATG8 proteins immobilized on glutathione beads. GST, GST-SUMO, and GST-Ub were used as controls. **(C)** Co-immunoprecipitation (co-IP) assays for assessment of the intracellular interactions between vIRF-1 and ATG8 proteins in virus-infected cells. iBCBL-1 cells were reactivated by treatment with 1 μg/ml doxycycline (Dox) for 2 days and fractionated into the nuclear and cytoplasmic fractions. The fractions were immunoprecipitated with normal rabbit IgG (nIgG) and rabbit anti-vIRF-1 antibody, and the immunoprecipitated complex was analyzed by immunoblotting with the indicated antibodies. Lamin B1 and lactate dehydrogenase (LDH) were used as nuclear and cytoplasmic fraction markers, respectively. **(D)** Immunofluorescence assay of the co-localization of vIRF-1 and GABARAPL1 in mitochondria (TOM20) in lytically virus-infected cells. iBCBL-1 cells were reactivated by treatment with Dox for 2 days and were fixed (Fix) with 4% paraformaldehyde before permeabilization (Per) with 0.5% Triton X-100 or after permeabilization with 25 μg/ml of saponin. A rat anti-vIRF-1 antibody was used. Scale bar, 10 μm. **(E)** Mutation of the core residues in the predicted LIR of vIRF-1: both tryptophan at position 255 (W255) and leucine at

position 258 (L258) were replaced with alanine. **(F-G)** GST pull-down assays for assessment of *in vitro* interactions of wild-type (WT) and A255/A258 vIRF-1 with GST-LC3B and GST-GABARAPL1. GST alone was used as a control. The whole-cell lysates of 293T cells expressing Flag-tagged vIRF-1 (vIRF-1-Flag) were used as input for the pull-down assay. **(G)** The intensities of co-precipitated vIRF-1 proteins relative to input were determined, and the data represent the mean ± SD of three independent experiments. 'ns', not significant. **(H)** Co-IP assay. 293T cells were co-transfected with vIRF-1(WT and A255/A258)-Flag with V5-LC3B or V5-GABARAPL1 and immunoprecipitated with anti-Flag antibody.

these results suggest that vIRF-1 exhibits a major feature of mitophagy receptors: direct interaction with ATG8 proteins.

To examine whether the predicted LIR ($^{254}$DWGRL$^{258}$) of vIRF-1 is required for ATG8 binding, the core residues tryptophan (W255) and leucine (L258) were replaced with alanine (Fig 1E). However, GST pull-down and co-IP assays showed that the vIRF-1 variant A255/A258 and native vIRF-1 were comparably co-precipitated with GABARAPL1 (Fig 1F–1H), indicating that the predicted LIR of vIRF-1 is not involved in the interactions with GABARAPL1. These results are consistent with another report showing that glycine (G) and arginine (R) between the core residues of LIR are not tolerated for interaction with ATG8 proteins [22].

## Identification of LIR-distinct GABARAPL1-interacting motif in vIRF-1

To identify the region of vIRF-1 required for GABARAPL1 binding, we used a series of successively refined deletion variants of vIRF-1 (Fig 2A), bacterially expressed as GST-fusion proteins for *in vitro* co-precipitation assays as previously described [23], together with purified recombinant T7-tagged GABARAPL1. GST pull-down assays showed that GST-fused full-length (FL) vIRF-1 and fragments containing the central region (amino acids 75–256 including the DBD) could precipitate T7-GABARAPL1, demonstrating the involvement of these sequences in the binding (Fig 2B). In line with this, a reverse pull-down assay showed that GST-GABARAPL1, but not GST alone and GST-LC3B, could co-precipitate the DBD-containing fragment of vIRF-1 (Fig 2C). The central region was successively deleted to map the core residues of vIRF-1 mediating GABARAPL1 binding, and the fragments were fused to the T7-intein-chitin binding domain for expression and detection (Fig 2D) as previously performed [23]. To this end, GST pull-down assays revealed that the C-terminal portion ($^{227}$SPGQCLPGEIVTPVPSCTTA$^{246}$) of the central region is involved in GABARAPL1 binding (Fig 2D). To provide further evidence that the 20 amino acids are essential for the binding in the context of full-length vIRF-1, we performed a co-IP assay with vIRF-1 full-length (FL) and its deletion variants Δ227–236 and Δ227–246 (Fig 2E). Indeed, V5-GABARAPL1 was not detected in the vIRF-1.Δ227-236-IP or Δ227-246-IP complex compared to the native vIRF-1-IP and NIX-IP complexes (Fig 2F). In addition, to further confirm the result, we performed a luciferase-based protein fragment complementation assay, NanoLuc Binary Technology (NanoBiT, Promega). NanoBiT is a two-subunit system, which is based on NanoLuciferase: Large BiT (LgB; 17.6 kDa) and Small BiT (SmB; 1.2 kDa) subunits fused to proteins of interest [24]. When the two proteins interact, the subunits come together to form an active enzyme and generate a bright luminescent signal in the presence of substrate. Flag-vIRF-1 (native and deletion variants) and HA-GABARAPL1 were fused to the C-terminus of SmB and LgB, respectively. LgB-LC3B and HaloTag-SmB were used as negative controls, and LgB-V5-NIX as a positive control for vIRF-1 binding. As expected, the luminescence intensity of a complex of vIRF-1 and GABARAPL1 was significantly reduced when the region 227–236 or 227–246 of vIRF-1 was deleted (Fig 2G). Together, these results suggest that region 227–236 is minimally needed for GABARAPL1 binding. Further study is warranted to identify single residue(s)

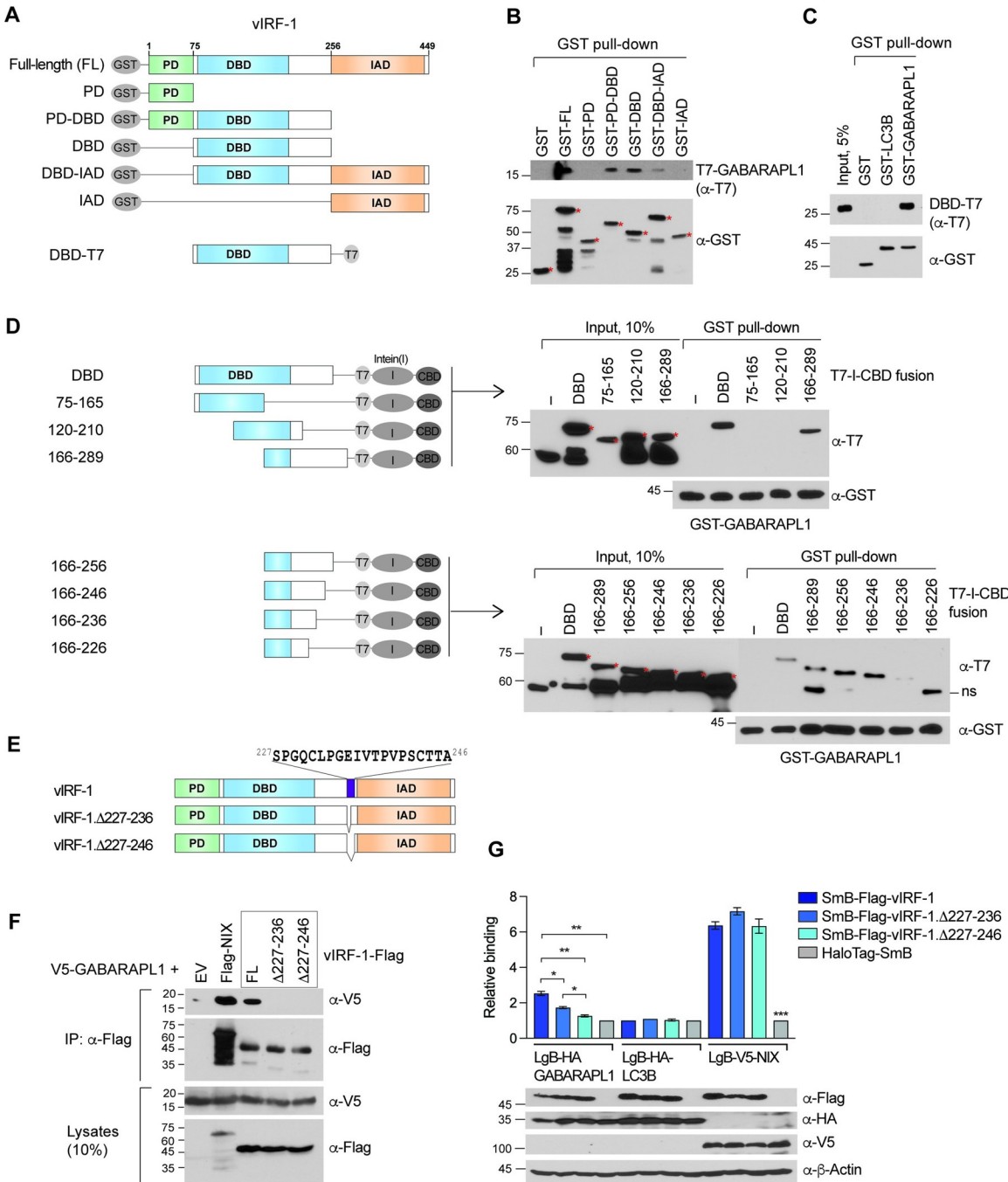

**Fig 2. vIRF-1 binds to GABARAPL1 via a short sequence motif in its central region. (A–C)** (A) Diagram of recombinant GST-vIRF-1 (progressively deleted) and vIRF-1 DBD-T7 proteins used in the GST pull-down assays in (B) and (C), respectively. Red asterisks indicate the fusion protein bands with the expected molecular weights. **(D)** GST pull-down assays with differently deleted vIRF-1 DBD polypeptides fused to T7-intein (I)-chitin-binding domain (CBD). The T7-I-CBD-fused polypeptides were expressed in bacteria, and whole-cell lysates were used for the *in vitro* binding assays. 'ns' indicates nonspecific bands. Red asterisks indicate the full-length T7-I-CBD-fused polypeptides. **(E)** Generation of the vIRF-1 variants (Δ227–236 and Δ227–246), in which the regions of S227 to I236 and S227 to A246 were deleted. **(F)** Co-IP assay. 293T cells were co-transfected with V5-GABARAPL1 with vIRF-1-Flag (WT and the variants) or Flag-NIX for 24 h. **(G)** NanoBiT assay was conducted in 293T cells transfected with the indicated NanoBiT binary (LgB and SmB) fusion proteins. Data are presented as the mean ± SD of three independent experiments. The one-way ANOVA test was used to assess the statistical significance of differences between groups, and the t-test was used for post hoc pairwise comparisons. *, $p < 0.05$; **, $p < 0.01$; and ***, $p < 0.001$. Immunoblots of the cell extracts are shown below the graph.

within the vIRF-1 region for GABARAPL1 binding. On the other hand, the NanoBiT interaction of NIX with vIRF-1 was not affected by the deletion mutation (Fig 2G).

## Identification of the single residues of GABARAPL1 required for vIRF-1 binding

Given that vIRF-1 binds to GABARAPL1 in an LIR-independent manner, we predicted that vIRF-1 might bind to a region other than the hydrophobic pockets HP1 and HP2 of GABARAPL1, which provide docking sites for the canonical LIR motif [25,26]. Thus, we sought to find the vIRF-1-interacting region of GABARAPL1 using a domain swap experiment. From the multiple sequence alignment of human ATG8 proteins using the web-based software Clustal Omega (S2A Fig), we defined seven blocks containing relatively poorly conserved 6 residues among ATG8 proteins and substituted each block in LC3B, which barely binds to vIRF-1 as shown in the previous protein interaction assays and is the most distal from GABARAPL1 in the phylogenetic tree (S2B Fig), with the counterparts of GABARAPL1 (Fig 3A). The resulting LC3B variants were fused to GST and expressed in bacteria for GST pull-down assays with purified vIRF-1-T7 protein. The results showed that the first or second block of GABARAPL1 conferred on LC3B the ability to bind to vIRF-1, albeit to a lesser extent than GABARAPL1 (Fig 3A). Subsequent mutagenesis studies were conducted to identify residues that might confer vIRF-1 binding. Firstly, each of the twelve residues within the first and second blocks of LC3B was substituted for the counterpart residues of GABARAPL1 (Fig 3B). GST pull-down assays revealed that three substitutions, D19E, L22K, and E41K among the twelve, conferred to LC3B the ability to co-precipitate vIRF-1 (Fig 3B). Secondly, to further verify the results, we substituted the residues E17, K20, and K38 of GABARAPL1 for the counterpart residues D19, L22, and E41 of LC3B (Fig 3C). As expected, all three mutations E17D/K20L/K38E and, to a lesser extent, double (E17D/K20L) and single (K38E) mutations, resulted in a loss of the binding activity of GABARAPL1 to vIRF-1 compared to WT (Fig 3C).

## vIRF-1 binds to GABARAPL1 in a manner different from p62/SQSTM1 and NIX

Based on the above findings, we reasoned that vIRF-1 binds to GABARAPL1 in a manner different from other autophagy and mitophagy receptors that interact with the hydrophobic pockets of ATG8 proteins. Thus, we sought to test the interactions of N-terminal-GABARAPL1 variants with the autophagy receptor p62/SQSTM1 compared to vIRF-1. Hereafter, we use the terms 'vIRF-1-interacting region (VIR)' for the region encompassing the residues E17, K20, and K38 of GABARAPL1 and 'VIR$^X$' for the variant E17D/K20L/K38E (Fig 4A). In addition to VIR$^X$, we generated the variant HP$^X$, in which the key residues K48, Y49, and L50 in the hydrophobic pockets of GABARAPL1 for LIR binding were replaced with alanine (K48A/Y49A/L50A) (Fig 4A). As expected, GST pull-down assays showed that the HP$^X$ variant, but not VIR$^X$, could bind to vIRF-1 (Fig 4B), thus firmly supporting the LIR-independent binding of vIRF-1 to GABARAPL1. In contrast, the autophagy receptor p62/SQSTM1, which plays a role as an autophagy receptor that recognizes and links ubiquitinated mitochondrial proteins to ATG8-containing phagophore membranes [27], could not bind to the HP$^X$ variant while it bound to WT and the VIR$^X$ (Fig 4C), indicative of the LIR-dependent binding of p62/SQSTM1 to GABARAPL1. In contrast to vIRF-1, p62/SQSTM1 could bind to LC3B (Fig 4C).

We next characterized the properties of intracellular interactions between GABARAPL1 and mitophagy receptors using co-IP assays. In addition to the VIR$^X$ and HP$^X$ variants, we included the GABARAPL1 variant G116A, in which glycine at position 116 was substituted for alanine, in the *in cellulo* interaction assays. The G116A variant is defective in cleavage for

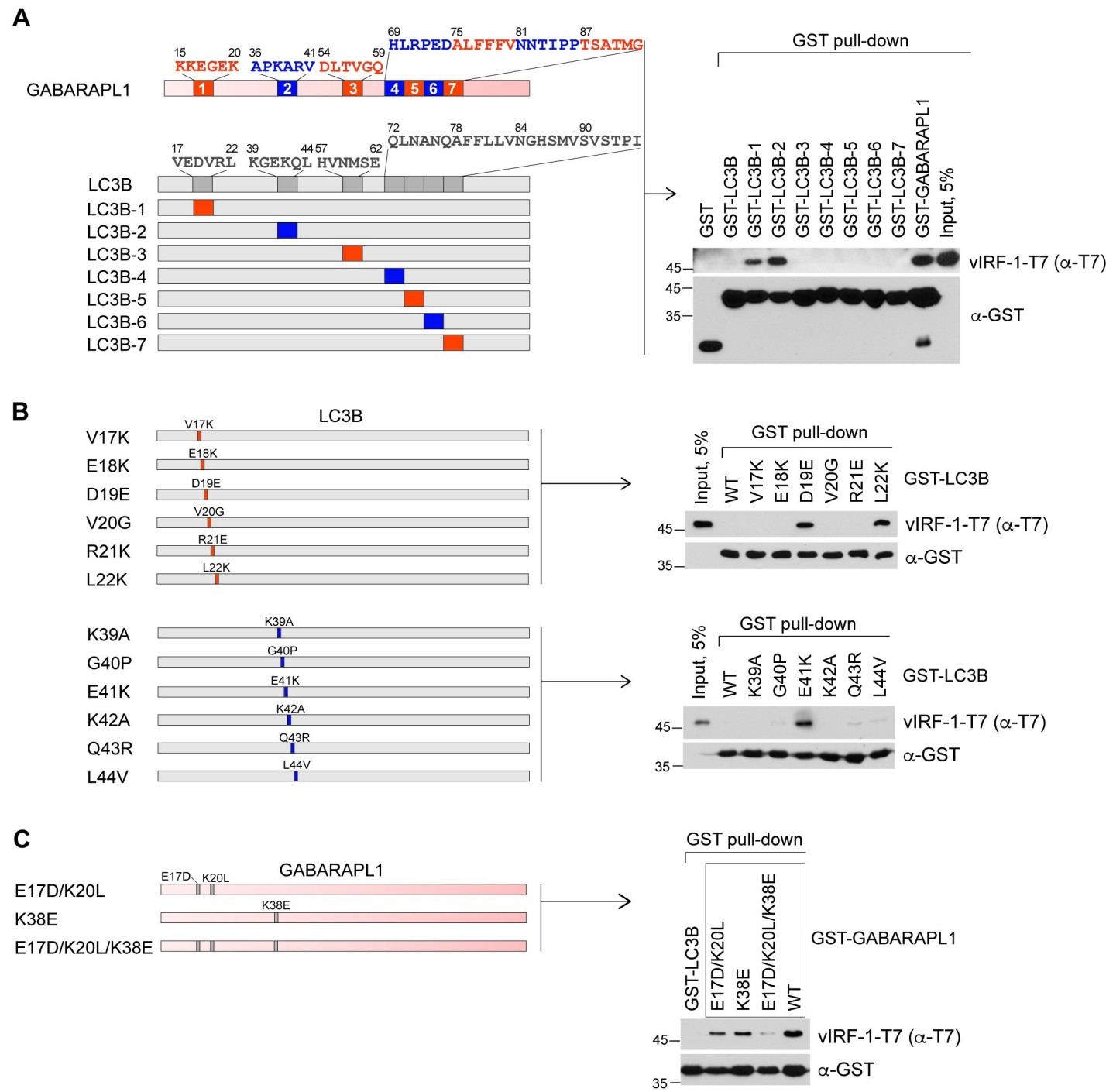

**Fig 3. Identification of single residues of GABARAPL1 required for vIRF-1 binding. (A-C)** GST pull-down assays were performed with purified vIRF-1-T7 and the following GST-fused LC3B or GABARAPL1 variant proteins: (A) LC3B-GABARAPL1 domain-swap variants in which hexameric blocks of diverged residues of LC3B were replaced with colinear residues of GABARAPL1; (B) LC3B point variants in which each residue in the first and second blocks of LC3B was replaced with the counterpart residue of GABARAPL1; (C) GABARAPL1 variants in which residues glutamic acid at 17 (E17), lysine residues at 20 (K20) and 38 (K38) of GABARAPL1 were replaced with the counterpart residues aspartic acid [D], leucine [L], and glutamic acid [E] of LC3B.

maturation, lipidation with phosphatidylethanolamine, and association with autophagosome membranes [28]. The binding assays showed that vIRF-1 and p62/SQSTM1 lost the ability to

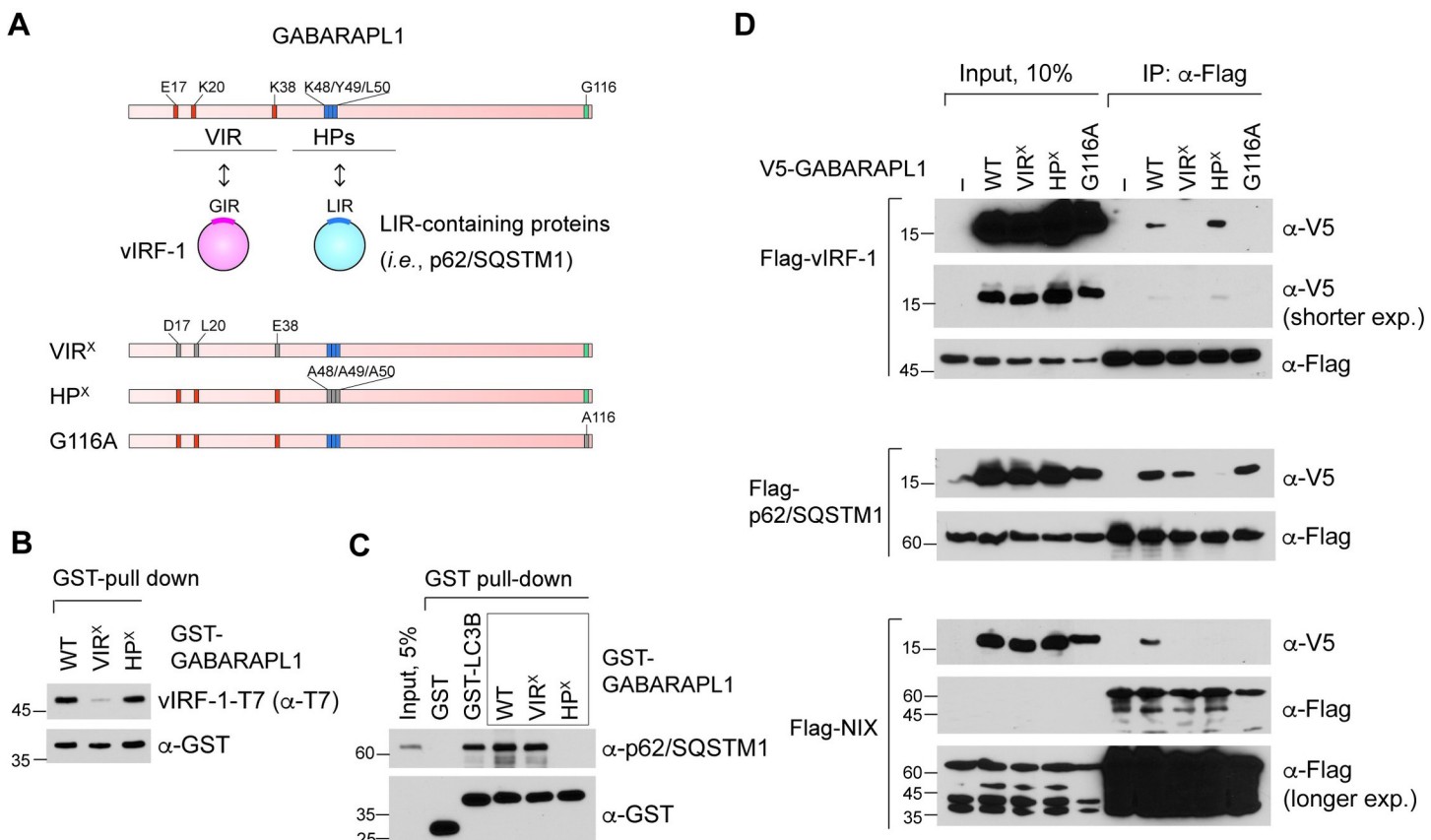

**Fig 4. vIRF-1 binds to GABARAPL1 in a manner different from cellular autophagy receptors. (A)** The schematic structure of GABARAPL1 and its variants, VIR[X], HP[X], and G116A, are defective in binding to vIRF-1, LIR-containing proteins, and in phosphatidylethanolamine-lipidation, respectively. GIR, GABARAPL1-interacting region; VIR, vIRF-1-interacting region; HP, hydrophobic pockets. **(B-C)** GST pull-down assays with the GST-fused GABARAPL1 variants and purified recombinant vIRF-1-T7 (B) and p62/SQSTM1 (C). **(D)** Co-IP assays for assessment of the intracellular interactions of the Flag-tagged mitophagy proteins vIRF-1, p62/SQSTM1, and NIX with V5-GABARAPL1 (WT and variants). 293T cells were transfected with the indicated proteins for 24 h. A longer exposure time was necessary to detect input Flag-NIX.

bind to GABARAPL1 when mutated in the VIR and the HP, respectively (Fig 4D), consistent with results from the *in vitro* binding assays (Fig 4B–4C). Intriguingly, vIRF-1 and p62/SQSTM1 exhibited distinct binding activities to the G116A variant; p62/SQSTM1, but not vIRF-1, could still bind to the variant. Together, these results suggest that p62/SQSTM1 can interact with the two different forms, soluble and phagosome membrane-anchored, of GABARAPL1, while vIRF-1 binds only to the membrane-anchored form. In addition, we examined the GABARAPL1 binding activity of another mitophagy receptor, NIX [29]. Surprisingly, the results showed that NIX could not bind to GABARAPL1 when either the VIR or the HP was mutated (Fig 4D), demonstrating that NIX requires both regions for GABARAPL1 binding. Furthermore, like vIRF-1, NIX lost the ability to bind to GABARAPL1 when G116 was mutated (Fig 4D). Together, these results suggest that vIRF-1 may represent a novel type of ATG8-binding protein that binds to the N-terminal region of the autophagosome membrane-anchored GABARAPL1 protein.

## The interaction between vIRF-1 and GABARAPL1 is involved in the regulation of mitochondria content and HHV-8 lytic replication

To examine the functional relevance of the interaction between vIRF-1 and GABARAPL1 in HHV-8 lytic replication, we generated an HHV-8 bacterial artificial chromosome 16 variant (BAC16.vIRF-1ΔGIR) encoding vIRF-1 lacking the GIR sequences (amino acids 227–236) using λ-Red recombination techniques described previously [30]. We verified the deletion of the targeted region and the integrity of the mutated versus wild-type BAC16 DNA by sequencing and gel electrophoresis after BspHI digestion (Fig 5A and 5B). BAC16 and BAC16.vIRF-1ΔGIR DNAs were stably infected into iSLK cells, which express Dox-inducible RTA and provide a tractable model cell line for HHV-8 infection [31]. We first investigated mitochondria content using immunoblotting of mitochondrially encoded NADH dehydrogenase subunit 1 (MT-ND1) in latent and lytically reactivated (Dox and sodium butyrate-treated) iSLK cells. The results showed that the level of MT-ND1 decreased in iSLK.BAC16 cells, but not iSLK. BAC16.vIRF-1ΔGIR, after reactivation (Fig 5C). Native and GIR-defective vIRF-1 proteins were comparably expressed after reactivation (Fig 5C). These results suggest that vIRF-1 interaction with GABARAPL1 plays a critical role in downregulating mitochondria content. Next, we examined virus productive replication in the iSLK cell lines. As expected, the production of encapsidated HHV-8 virions significantly decreased in reactivated iSLK.BAC16.vIRF-1ΔGIR cells compared to reactivated iSLK.BAC16 cells (Fig 5D).

Next, we sought to investigate the role of GABARAPL1 in HHV-8 productive replication. We generated control and *GABARAPL1* knockout (KO) iSLK.BAC16 cells using CRISPR/Cas9 and verified the loss of GABARAPL1 using immunoblotting (Fig 5E). Consistent with the result in reactivated iBCBL-1 (S1 Fig), the expression of GABARAPL1 increased in iSLK. BAC16 cells after reactivation (Fig 5E). We next performed HHV-8 replication assays in control and *GABARAPL1* KO iSLK.BAC16 cells. The results showed that productive virus replication was significantly decreased by GABARAPL1 deficiency but rescued by reconstitution with sgRNA-refractory GABARAPL1 but not the mutant VIR[X] (Fig 5F). Moreover, the immunoblotting analysis showed that mitochondria content (MT-ND1) was significantly reduced by reconstitution with native GABARAPL1, but not the mutant VIR[X], only in reactivated control and *GABARAPL1* KO iSLK.BAC16 cells (Fig 5F), indicative of the demand of vIRF-1 interaction with GABARAPL1 for autophagic clearance of mitochondria and lytic replication. These results suggest that the interaction between vIRF-1 and GABARAPL1 is required for reactivation-induced mitochondria clearance and productive virus replication.

## The interaction between vIRF-1 and GABARAPL1 contributes to mitophagy

To assess if the vIRF-1:GABARAPL1 interaction contributes to the promotion of mitophagy flux, we used the HeLa.Kyoto cell line that stably expresses the mitophagy reporter 'mito-mCE', previously described [18], in which the mCherry-EGFP tandem fluorescence proteins were fused to TOM20 N-terminal residues 1–33 for mitochondrial targeting (Fig 6A). The EGFP signal is quenched under acidic pH conditions, while mCherry can be visualized at lower pH; therefore, the fusion of mitochondrion-containing autophagosomes with lysosomes causes the loss of yellow fluorescence and the appearance of only red fluorescence of mCherry, indicative of the terminal step in mitophagy [32]. The ratio of red to green fluorescence was determined using FIJI software. Note that leupeptin, an inhibitor of lysosomal proteases, was added to cultures to facilitate the detection of end-stage mitophagy [17]. As a positive control for mitophagy activation, we co-expressed vIRF-1 and NIX, previously shown to lead to a considerable decrease in mitochondria content in HeLa.Kyoto cells [17]. Indeed, co-transfection

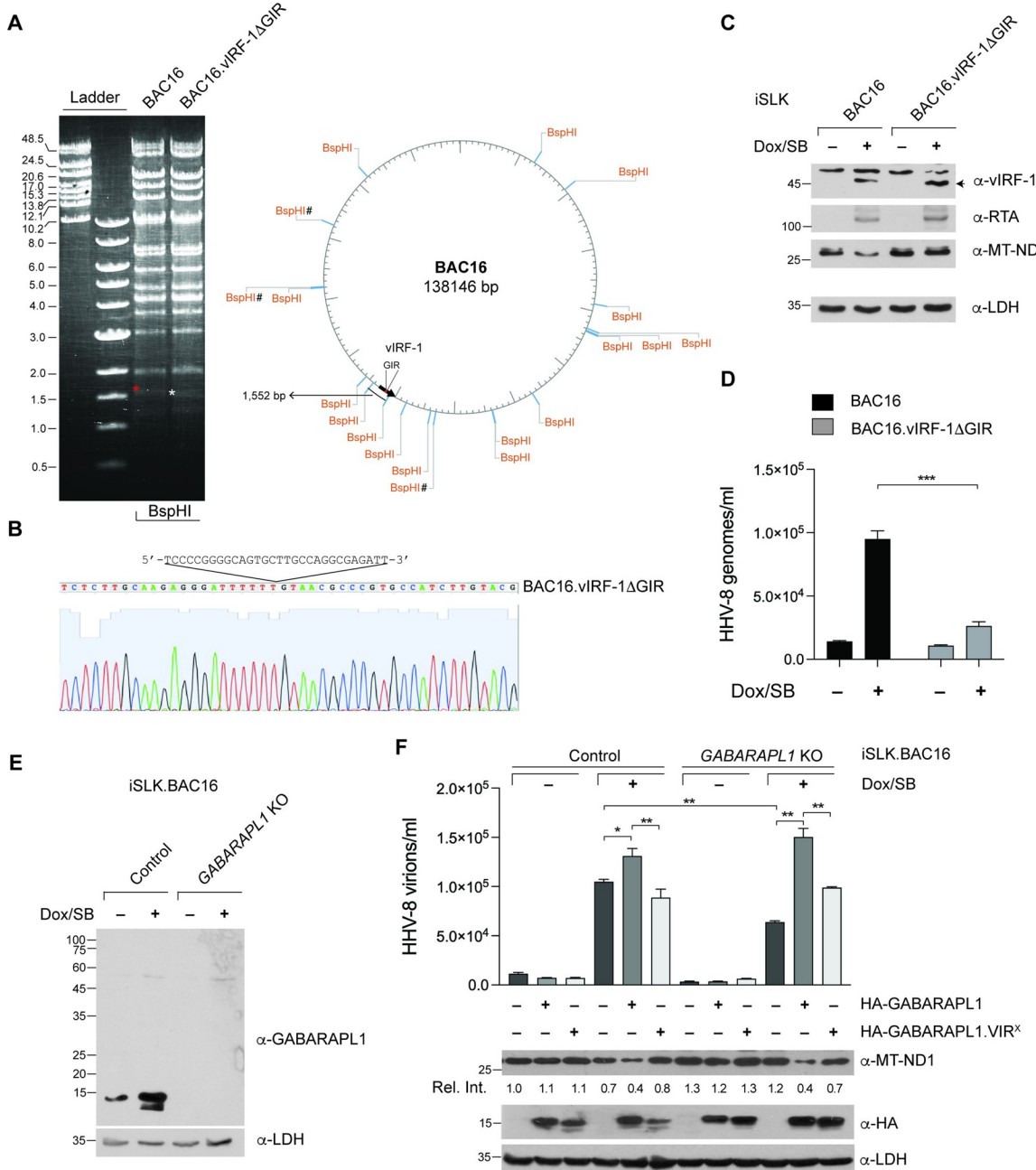

**Fig 5. The interaction between vIRF-1 and GABARAPL1 is involved in regulating mitochondria content and HHV-8 productive replication.** (A) Agarose gel electrophoresis of the BspHI-digested DNA fragments of BAC16 and the mutant BAC16.vIRF-1ΔGIR (227–236) genomes. The DNA fragments (1,552 and 1,522 base pairs), including and deficient in the vIRF-1 GIR, are indicated with red and white asterisks, respectively. The BAC16 genome is drawn using NEBcutter V3.0. Sharp signs indicate methylation sites that may affect BspHI digestion. (B) DNA sequence verification of the deletion mutation of the vIRF-1 GIR sequences, as noted in the BAC16. vIRF-1ΔGIR genome. (C) Immunoblot analysis of extracts derived from iSLK.BAC16 and iSLK.BAC16.vIRF-1ΔGIR left untreated or treated with Dox and sodium butyrate (SB) for 3 days. MT-ND1 indicates mitochondrially-encoded NADH dehydrogenase 1. (D) RT-qPCR analysis of the encapsidated viral genome copy number in the media of the above iSLK cell cultures. Data represent the mean ± SD of three independent experiments. *** < 0.001. (E) Immunoblot verification of CRISPR/Cas9-mediated knockout (KO) of GABARAPL1 in iSLK.BAC16 cells. Control cells were generated by transducing the parental lentiCRISPR v2 vector. (F) Determination of the HHV-8 genome copy number in the media of control and *GABARAPL1* KO iSLK.BAC16 cells left untreated or treated with Dox and SB for 3 days. The corresponding cell extracts were immunoblotted with the indicated antibodies. The relative band intensities of MT-ND1 normalized to the loading control LDH are displayed beneath the corresponding panel. The one-way ANOVA test was used to assess the statistical significance of differences between groups, and the t-test was used for post hoc pairwise comparisons. *, $p < 0.05$ and **, $p < 0.01$.

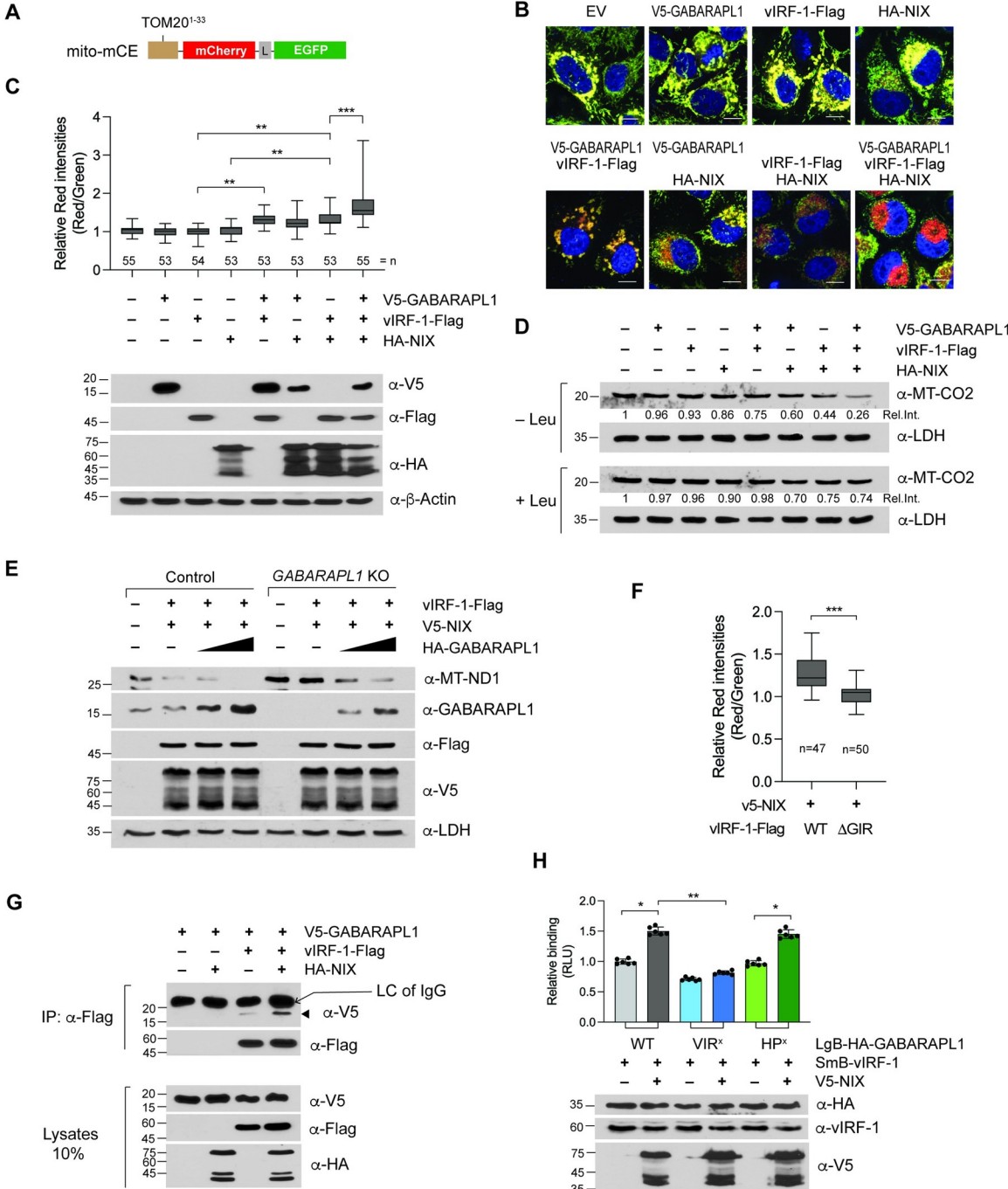

**Fig 6. GABARAPL1 is required for vIRF-1/NIX-mediated mitophagy. (A)** Diagram of the mitophagy reporter, mito-mCE. The mCherry-EGFP tandem tag was fused to the mitochondrial targeting signal sequences (amino acids 1 to 33) of TOM20. A glycine-serine linker (L) was introduced between mCherry and EGFP. **(B-C)** The HeLa.Kyoto cell line (HeLa.Kyoto^mito-mCE) stably expressing mito-mCE was generated and then transiently transfected with empty vector (EV) or plasmids expressing vIRF-1, GABARAPL1, and NIX for 24 h in the presence of 40 μM leupeptin. (B) Confocal images of the transfected HeLa.Kyoto^mito-mCE cells. Scale bar, 10 μm. (C) Quantification of mitophagy. When the two microscopic images of EGFP and mCherry (see S2 Fig) were merged, the cells showing a stronger red fluorescence were regarded as positive for mitophagy. The red-to-green fluorescence ratio of more than 50 cells showing mitophagy from ten randomly collected images was determined using the ImageJ (Fiji) software [46]. The cell number (n) counted is noted under the graph column. Cells were cultured in parallel to examine the expression levels of the transfected genes by immunoblot analysis. The one-way ANOVA test was used to assess the statistical significance of differences between groups, and the t-test was used for post hoc comparisons. **, $p < 0.01$ and ***, $p < 0.001$. **(D)** Immunoblots of lysates derived from HeLa.Kyoto cells transfected with plasmids expressing vIRF-1-Flag, V5-GABARAPL1, and HA-NIX for 24 h in the presence and absence of 40 μM leupeptin. The relative

intensities (Rel. Int.) of MT-CO2 bands were determined by dividing them with that in empty vector control and noted under each band. **(E)** Immunoblot analysis of extracts of control and *GABARAPL1* KO HeLa.Kyoto cells transfected with plasmids expressing vIRF-1-Flag (500 ng), V5-NIX (200 ng), and HA-GABARAPL1 (50 and 200 ng) for 24 h. **(F)** Mitophagy assay in HeLa.Kyoto$^{mito-mCE}$ cells co-transfected with V5-NIX together with WT or Δ227–236 vIRF-1. 'n' indicates the number of cells counted. ***, $p < 0.001$. **(G-H)** Co-IP and NanoBiT assays for assessment of the effects of NIX expression on the interaction between vIRF-1 and GABARAPL1. 293T cells were transfected with the indicated plasmids for 24 h. **(G)** Immunoblot analysis of cell lysates and Flag-IP complexes derived from the transfected cells. Arrowhead indicates the band of V5-GABARAPL1 co-precipitated with vIRF-1, and 'LC' indicates the light chain of immunoglobulin G (IgG). **(H)** The NanoBiT data are presented as the mean ± SD of six independent wells per condition. The one-way ANOVA test was used to assess the statistical significance of differences between groups, and the t-test was used for post hoc pairwise comparisons. **, $p < 0.01$ and *, $p < 0.05$. Immunoblots of the cell extracts are shown below the chart.

of vIRF-1 and NIX induced a higher ratio of red to green fluorescence along with increased levels of red dots at the perinuclear area compared to empty vector (EV) control and single transfection of vIRF-1 or NIX (Figs 6B–6C and S3). We speculated that the protein expression level of endogenous NIX in HeLa.Kyoto cells may be too low to activate vIRF-1-mediated mitophagy. Indeed, NIX was barely detected in the cell line, whereas it was readily detected in iBCBL-1 cells (S4 Fig).

Subsequent experiments revealed that the co-transfection of vIRF-1 and GABARAPL1 could induce a greater red-to-green fluorescence ratio compared to each single or empty vector control transfection (Fig 6B–6C). Surprisingly, co-transfection of GABARAPL1 with both vIRF-1 and NIX led to a much higher red-to-green fluorescence (Fig 6B–6C). To confirm these results, we examined changes in mitochondria content in the above cell culture conditions, but without leupeptin, by immunoblotting MT-CO2 (*mt*DNA-encoded *c*ytochrome c *o*xidase *II*). As expected, the results showed that GABARAPL1 could promote a decrease in mitochondria content when co-transfected with either NIX or vIRF-1 and, to a greater extent, with both NIX and vIRF-1 (Fig 6D). However, adding leupeptin to the cell cultures blocked the decrease in mitochondria content (Fig 6D). These findings suggest that GABARAPL1 is involved in the acceleration of vIRF-1/NIX-induced mitophagy flux and mitochondria clearance. We next examined, using *GABARAPL1* KO HeLa.Kyoto cells generated by CRISPR/Cas9, whether GABARAPL1 is required for vIRF-1/NIX-induced mitophagy. We first examined mitochondria content (MT-ND1) in transfected control and *GABARAPL1* KO HeLa.Kyoto cells. Immunoblotting analysis showed that a decrease in MT-ND1 induced by vIRF-1/NIX expression was inhibited in *GABARAPL1* KO HeLa.Kyoto cells but restored by reconstitution with sgRNA-refractory GABARAPL1 in a dose-dependent manner (Fig 6E). In line with this, mitophagy flux assays showed that vIRF-1/NIX-induced mitophagy was significantly inhibited in the *GABARAPL1* KO cells, and the reduced mitophagy activity was rescued by complementation of GABARAPL1, but not VIR$^X$, HP$^X$, or G116A (S5 Fig). Either vIRF-1 or NIX could activate mitophagy, to a lesser extent than vIRF-1/NIX, when co-transfected with GABARAPL1 WT, but not VIR$^X$, HP$^X$, or G116A, into *GABARAPL1* KO HeLa.Kyoto cells (S5 Fig). Surprisingly, vIRF-1 could not induce mitophagy when co-transfected with the HP$^X$ variant of GABARAPL1 (S5 Fig). This may imply that an LIR motif-containing autophagy factor, such as NIX, is recruited to the vIRF-1/GABARAPL1 complex to promote mitophagy. Furthermore, the vIRF-1ΔGIR variant could not activate mitophagy when co-transfected with NIX (Fig 6F). Together, these results suggest that GABARAPL1 is required for mitophagy activation by the vIRF-1/NIX complex.

## NIX promotes vIRF-1 interaction with GABARAPL1

We next investigated whether the cellular mitophagy receptor NIX affects the interaction of vIRF-1 with GABARAPL1 using a co-IP assay. The results showed that NIX expression promoted the interaction of GABARAPL1 with vIRF-1 (Fig 6G). Furthermore, a NanoBiT binary

interaction between vIRF-1 and GABARAPL1 WT and HP$^X$, but not VIR$^X$, was significantly enhanced by NIX expression (Fig 6H). These results suggest that NIX contributes to vIRF-1-activated mitophagy by promoting interaction between vIRF-1 and GABARAPL1.

## NIX induces the dimerization and aggregation of cytoplasmic vIRF-1

According to the Human Protein Atlas and Uniprot servers [33,34], NIX can be localized to the nucleus in addition to mitochondria. vIRF-1 can also localize to the nucleus [23,35]. Thus, to rule out a possible role of the nuclear-localized proteins in mitophagy activation, we employed a nuclear localization signal (NLS)-mutated version of vIRF-1 (NLS$^X$) (Fig 7A), shown previously to be inactive in the inhibition of p53-, SMAD3-, or IRF3-mediated transactivation [16,23]. The result showed that co-expression of NIX and vIRF-1 NLS$^X$ led to a much higher red-to-green fluorescence ratio than that of NIX and WT vIRF-1 (Fig 7B). Considering that the total expression levels of the vIRF-1 WT and the variant NLS$^X$ are comparable (Fig 7C), this might be due to the relatively high mitochondria-localized vIRF-1 NLS$^X$. Interestingly, we could detect the presence of slower migrating bands of vIRF-1 NLS$^X$ and, to a lesser extent, vIRF-1 WT in the extracts from cells co-transfected with NIX (Fig 7C). The bands may represent covalently modified and/or aggregated proteins. Thus, we examined using IFA whether vIRF-1 is detected as an aggregate in the mitochondria of NIX-transfected cells. In addition to standard protocol, cells were pre-permeabilized with saponin before fixation to facilitate the detection of mitochondria-localized vIRF-1 by removing free cytosolic vIRF-1. Overexpression of NIX could promote mitochondria targeting of vIRF-1 WT and NLS$^X$ but induce the formation of NLS$^X$ speckles with a bigger size than that of WT (Fig 7D). However, the NLS$^X$ speckles appeared not to be co-localized with mitochondria but in the vicinity of mitochondria together with NIX (Fig 7D). While it is uncertain if the NLS$^X$ speckles represent aggregated forms of vIRF-1 or if there is a technical issue in IFA detection of the NLS$^X$ speckles in mitochondria, we wanted to examine whether an aggregated form of the NLS$^X$ is detected in isolated mitochondria. Immunoblotting analysis showed that the NLS$^X$ could indeed be detected as aggregated forms in the enriched mitochondria fraction when co-transfected with NIX. (Fig 7E). Furthermore, NanoBiT assays showed that the NLS$^X$ could significantly bind to GABARAPL1 compared to WT vIRF-1 when co-transfected with NIX (Fig 7F). Together, these results suggest that NIX-induced aggregation of cytoplasmic vIRF-1 may be functionally related to mitophagy activation.

We next examined whether vIRF-1 with higher molecular weights (>140 kDa) can be detected in reactivated iBCBL-1 cells. We used detergent soluble and insoluble (DRM) fractions. Indeed, vIRF-1 proteins with high molecular weights were readily detected in the DRM fraction of lytic iBCBL-1 cells after Dox treatment for 2 and 3 days (Fig 8A). Interestingly, lytic reactivation increased the expression of dimeric NIX, which is a mitophagy-competent form [17,36], in the DRM fraction (Fig 8A). The increased expression of dimeric NIX might be caused by the migration of the dimeric NIX from the detergent-soluble fraction to the DRM fraction, as the levels of soluble dimeric NIX proteins were reduced following reactivation (Fig 8A). Note that we previously demonstrated that the mRNA expression of NIX is not affected by reactivation in iBCBL1 cells [17]. To investigate whether NIX is required for lytic reactivation-induced vIRF-1 aggregation, we generated *NIX* KO iBCBL-1 cells using CRISPR/Cas9. Indeed, the level of vIRF-1 proteins with molecular weights of more than 140 kDa was significantly reduced in the lytic *NIX* KO cells compared to control cells (Fig 8B). Notably, the levels of autophagy-competent TUFM, a dimerized form induced by vIRF-1 expression [18], were reduced in *NIX* KO iBCBL-1 cells (Fig 8B). These results suggest that NIX is upstream of vIRF-1-mediated mitophagy via TUFM in lytically HHV-8-infected cells. In line with this, the

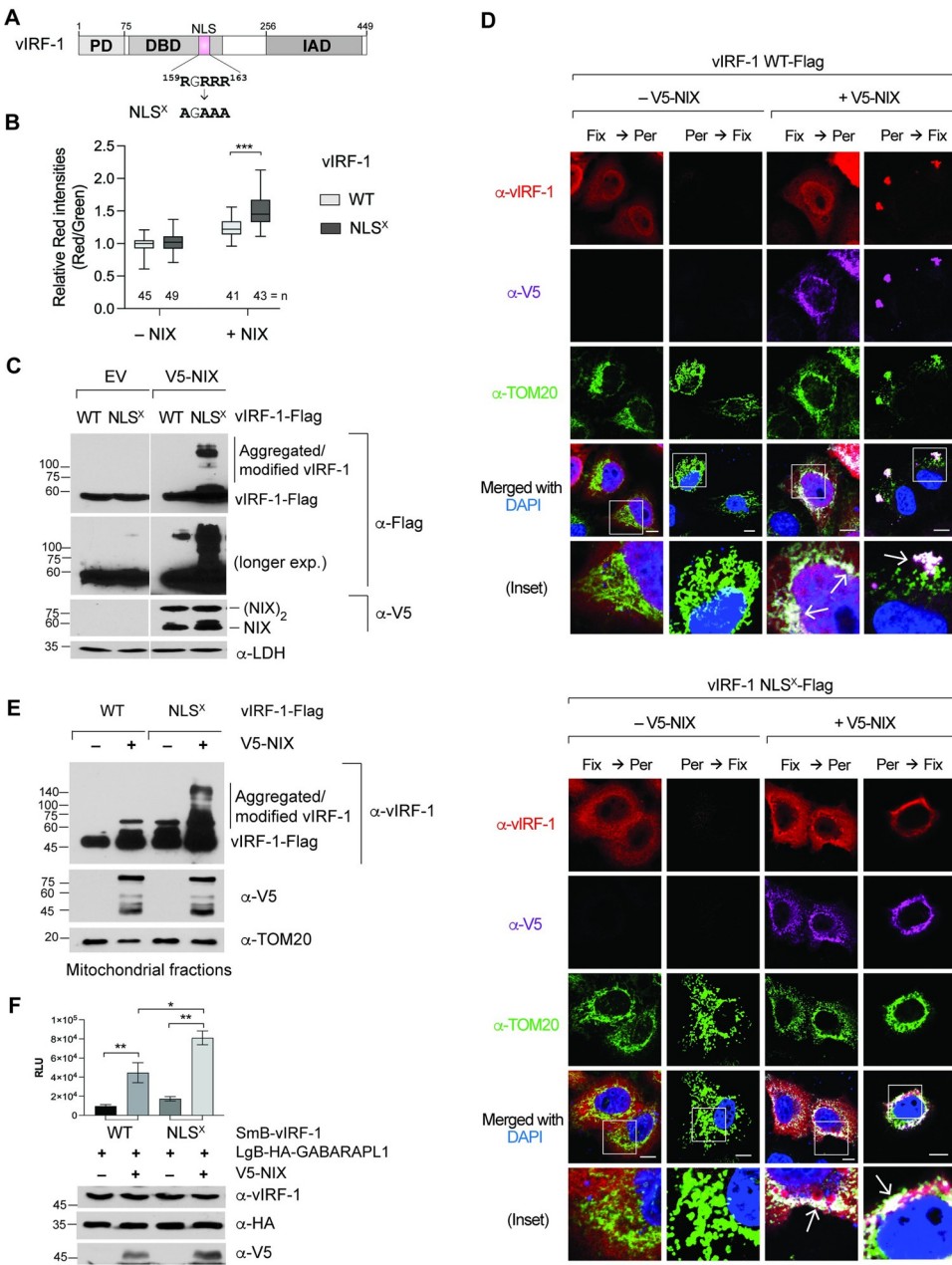

**Fig 7. NIX promotes the generation of vIRF-1 proteins with higher molecular weight. (A)** vIRF-1's nuclear localization signal (NLS) and its mutation (NLS$^X$). **(B-C)** Mitophagy (B) and immunoblot (C) analyses in HeLa. Kyoto$^{mito-mCE}$ cells transfected with WT or NLS$^X$ vIRF-1 vector along with or without V5-NIX for 24 h. **(D)** IFA analysis of vIRF-1 and vIRF-1.NLS$^X$ in HeLa.Kyoto cells co-transfected with or without V5-NIX plasmid. Cells were fixed (Fix) before or after permeabilization (Per). Arrows indicate the co-localization of vIRF-1 and NIX in mitochondria. Scale bar, 10 μm. **(E)** Immunoblot analysis of the mitochondrial extracts derived from HeLa.Kyoto cells transfected with the indicated plasmids. **(F)** NanoBiT assay. 293T cells were transfected with the indicated NanoBiT plasmids with or without V5-NIX. Data are presented as the mean ± SD of six independent wells per condition. The one-way ANOVA test assessed the statistical significance of differences between groups, and the t-test was used for post hoc pairwise comparisons. **, $p < 0.01$ and *, $p < 0.05$. Immunoblots of the cell extracts are shown below the chart.

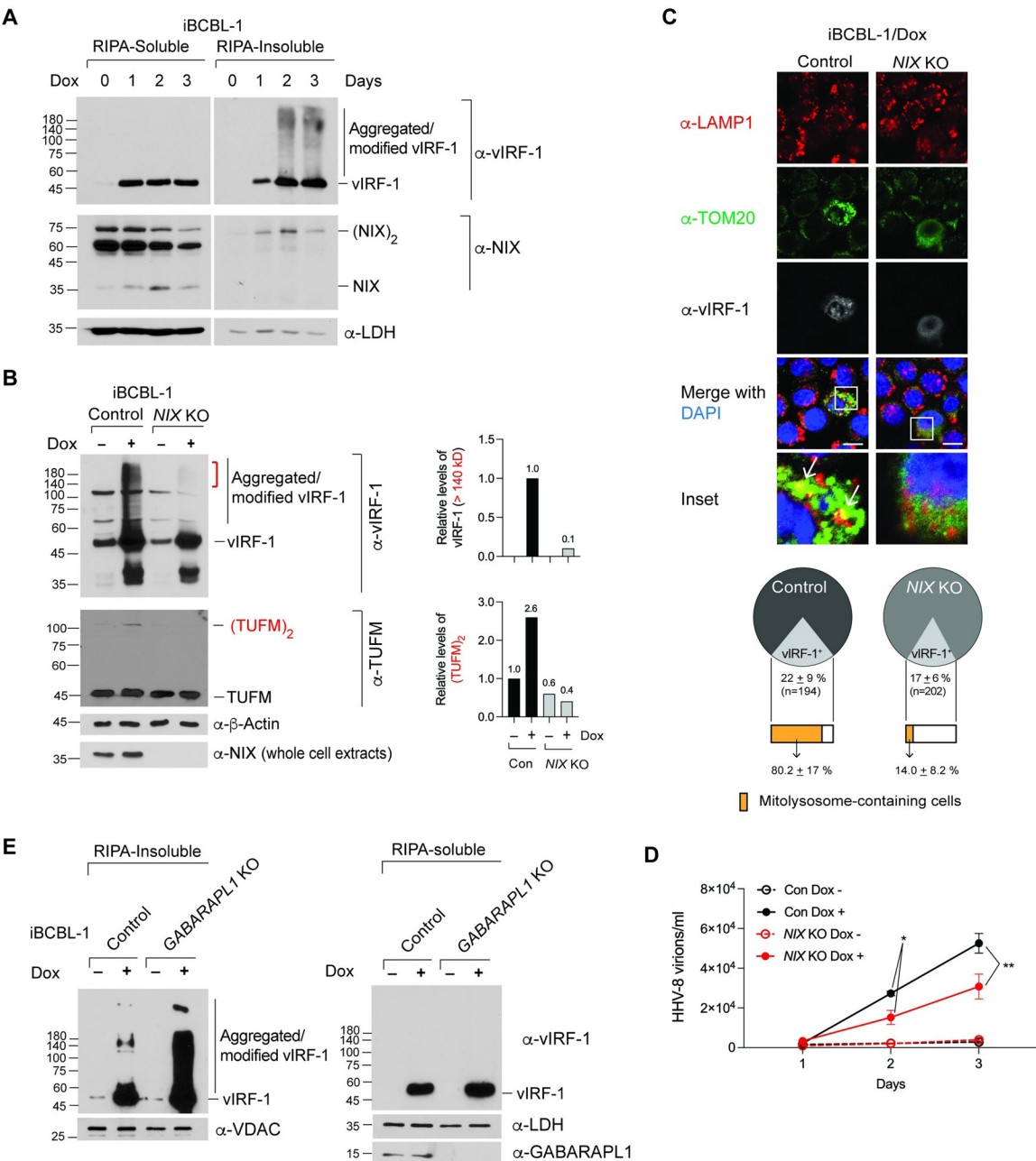

**Fig 8. NIX, but not GABARAPL1, promotes vIRF-1 aggregation for reactivation-induced mitophagy and productive virus replication. (A)** Immunoblot analysis of the RIPA-soluble and insoluble (DRM) fractions derived from iBCBL-1 cells treated with Dox for 0 to 3 days. (NIX)$_2$ indicates dimerized NIX. **(B)** Immunoblot analysis of the DRM fraction derived from control and *NIX* KO iBCBL-1 cells treated without or with Dox for 2 days. (TUFM)$_2$ indicates dimerized TUFM, and the red line indicates vIRF-1 proteins with a mass of more than 140 kDa. Relative band intensities of vIRF-1 (> 140 kDa) and (TUFM)$_2$ are graphed. For NIX immunoblotting, whole-cell extracts were used. **(C)** IFA of control and *NIX* KO iBCBL-1 cells reactivated by Dox treatment for 2 days. Leupeptin was added to the culture for 24 h before fixation. Arrows indicate autolysosomes containing mitochondria (termed mitolysosomes), the co-localization of LAMP1 and TOM20. The percentages of vIRF-1-expressing cells and mitolysosome-containing cells are shown in the charts. A rat anti-vIRF-1 antibody was used for the triple-color IFA. Scale bar, 10 μm. **(D)** RT-qPCR analysis of the copy number of the encapsidated viral genome present in the culture media of control and *NIX* KO iBCBL-1 cells left untreated or treated with Dox for 1 to 3 days. Data represent the mean ± SD of three independent experiments. The one-way ANOVA test was used to assess the statistical significance of differences between groups, and the t-test was used for post hoc pairwise comparisons. ** $p < 0.01$ and * $p < 0.05$. **(E)** Immunoblot analysis of the RIPA-soluble and insoluble fractions derived from control and *GABARAPL1* KO iBCBL-1 cells left untreated or treated with Dox for 2 days. VDAC was used as a marker of the DRM fraction.

formation of mitochondria-containing lysosomes in vIRF-1-expressing (lytic) iBCBL-1 cells was significantly reduced by NIX deficiency (Fig 8C). Furthermore, the genome copy number of encapsidated HHV-8 virions was greatly reduced in *NIX* KO iBCBL-1 cells compared to control cells (Fig 8D), suggesting NIX-mediated mitophagy is essential for virus productive replication.

### Loss of GABARAPL1 promotes vIRF-1 aggregation

On the other hand, we were curious about the role of GABARAPL1 in reactivation-induced vIRF-1 aggregation. Surprisingly, the loss of GABARAPL1 led to a robust increase in reactivation-induced vIRF-1 modifications and aggregation in the DRM fraction but not in the soluble fraction of iBCBL-1 cells (Fig 8E). These results might be attributed to an accumulation of vIRF-1-containing mitochondria induced by mitophagy inhibition in GABARAPL1-deficient cells.

### NIX promotes vIRF-1 dimerization and increases the stability of aggregated vIRF-1

We next wondered how NIX could promote vIRF-1 aggregation. NanoBiT assays showed that NIX could promote the dimeric interaction of vIRF-1 WT and, to a greater extent, for vIRF-1 NLS$^X$ (Fig 9A). However, NIX.ΔTA, which lacks the tail-anchor (TA) domain for mitochondrial targeting, could not promote the dimerization of vIRF-1 (Fig 9A). The TA domain has a GXXXG motif responsible for NIX's dimerization on the mitochondrial membrane [36]. Thus, it is likely also that NIX dimerization may be required for vIRF-1 dimerization. These results suggest that NIX-induced vIRF-1 dimerization might be involved in producing the higher molecular weight bands of vIRF-1 in mitochondria.

In addition, to determine whether NIX can stabilize the higher molecular weight form of vIRF-1, HeLa.Kyoto cells were co-transfected with vIRF-1 along with or without NIX, and cells were subjected to cycloheximide (CHX) chase assay over 8 h. The results showed that high molecular weight vIRF-1 proteins, including the aggregated form, remained detectable without any loss until 4 h post-CHX treatment in the DRM fraction of NIX-expressing cells (Fig 9B–9C). In contrast, the vIRF-1 proteins with a more than 140 kDa mass were barely detected after CHX treatment in the DRM fraction of control empty vector cells (Fig 9B–9C). Interestingly, the levels of the vIRF-1 protein with a mass of 50 kDa were not changed in the detergent soluble and DRM fractions (Fig 9B–9C). These results suggest that modified and aggregated vIRF-1 exhibit a higher protein turnover than unmodified vIRF-1. Notably, the NIX protein's expression was coincidently downregulated in the DRM fraction after CHX treatment (Fig 9B), consistent with NIX's possible role in stabilizing high molecular weight vIRF-1 proteins. Altogether, our findings suggest that induced expression of NIX may promote the generation of mitophagy-competent vIRF-1.

### Discussion

Our prior and current studies suggest that the mutual interaction between vIRF-1 and NIX promotes mitophagy during lytic replication (Fig 10). Furthermore, the complex direct and indirect interactions of mitochondria-localized vIRF-1 with the autophagy machinery may facilitate the process of autophagy, including initiation, elongation, and complete closure of autophagosome membranes (see below). Further studies are warranted to identify the exact molecular mechanism by which NIX induces vIRF-1 modifications and aggregation and to reveal how the modifications of vIRF-1 are related to GABARAPL1 binding and mitophagy activation.

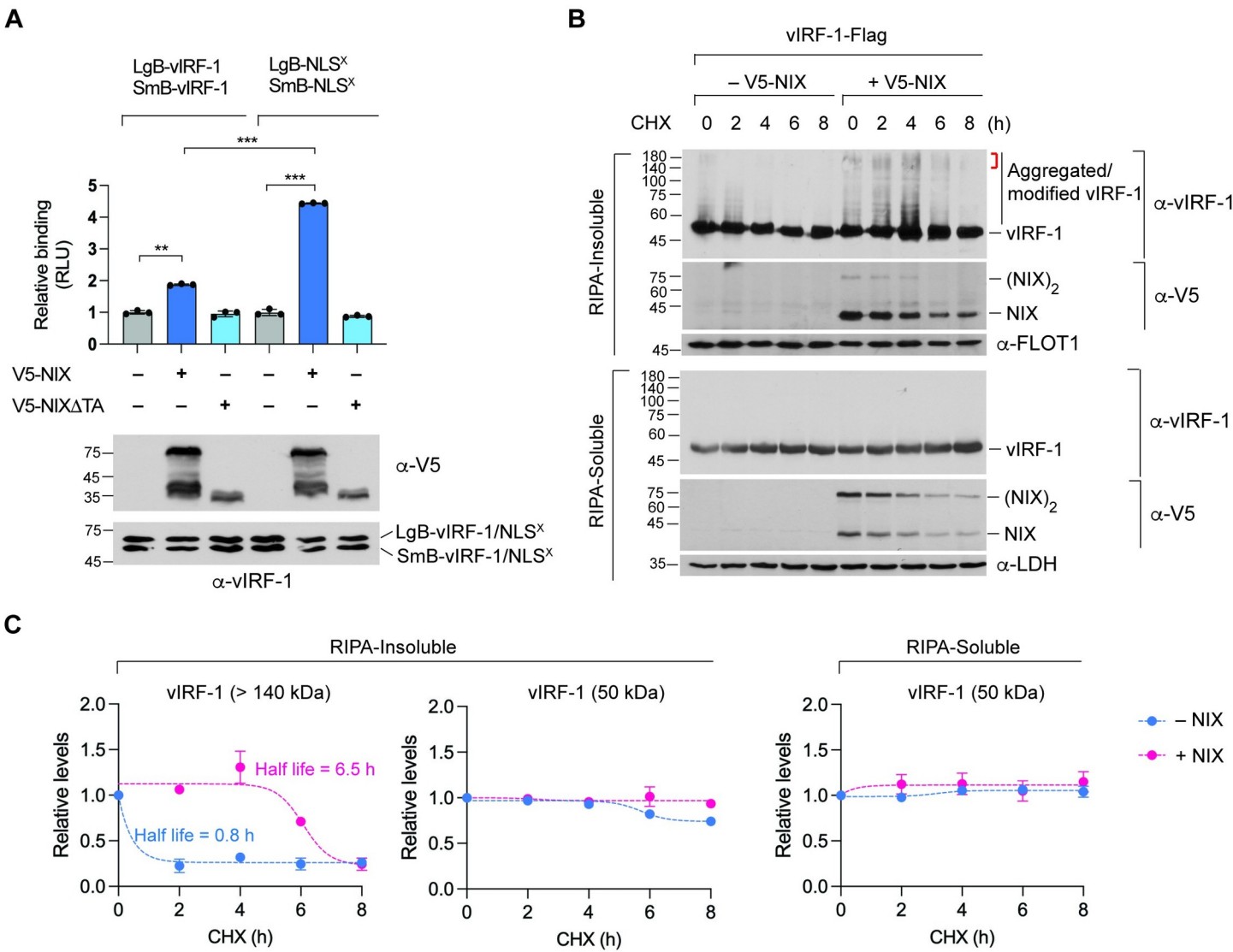

**Fig 9. NIX promotes vIRF-1 dimerization and stabilizes aggregated vIRF-1. (A)** NanoBiT assays of NIX-promoted vIRF-1 dimerization. Data are presented as the mean ± SD of three independent wells per condition. The one-way ANOVA test was used to assess the statistical significance of differences between groups, and the t-test was used for post hoc pairwise comparisons. ***, $p < 0.001$ and **, $p < 0.01$. The expression levels of the NanoBiT vIRF-1 (WT and NLS$^X$) and V5-NIX (WT and ΔTA) proteins were examined using immunoblot analysis. **(B)** Immunoblot analysis of the RIPA-soluble and insoluble fractions derived from HeLa.Kyoto cells co-transfected with vIRF-1 with or without V5-NIX for 24 h and then treated with cycloheximide (CHX) for 0, 2, 4, 6, and 8 h. The red line indicates vIRF-1 proteins with a more than 140 kDa mass. Representative images are shown from three independent experiments. **(C)** Relative levels of monomeric vIRF-1 (50 kDa) and vIRF-1 aggregates above 140 kDa detected in (B) were quantified and indicated relative to levels at the initiation (0 h) of CHX treatment in the plots. Half-life was determined using a non-linear regression fit.

Mitophagy receptors are involved in the execution of mitophagy by connecting cargo mitochondria to the autophagosomal membranes via interaction with ATG8 family proteins [37]. Therefore, mitophagy receptor proteins have to meet at least three criteria: 1) localization to mitochondria, 2) interaction with ATG8 proteins in response to a stimulus, and 3) possession of a short linear motif, such as LIR, for direct interaction with ATG8 proteins. We previously demonstrated that vIRF-1 could be localized, in part, to mitochondria by targeting detergent-resistant microdomains on the OMM [16]. The present study also found that vIRF-1 can interact directly with ATG8 family proteins. Interestingly, vIRF-1 binds selectively to GABARAPL1

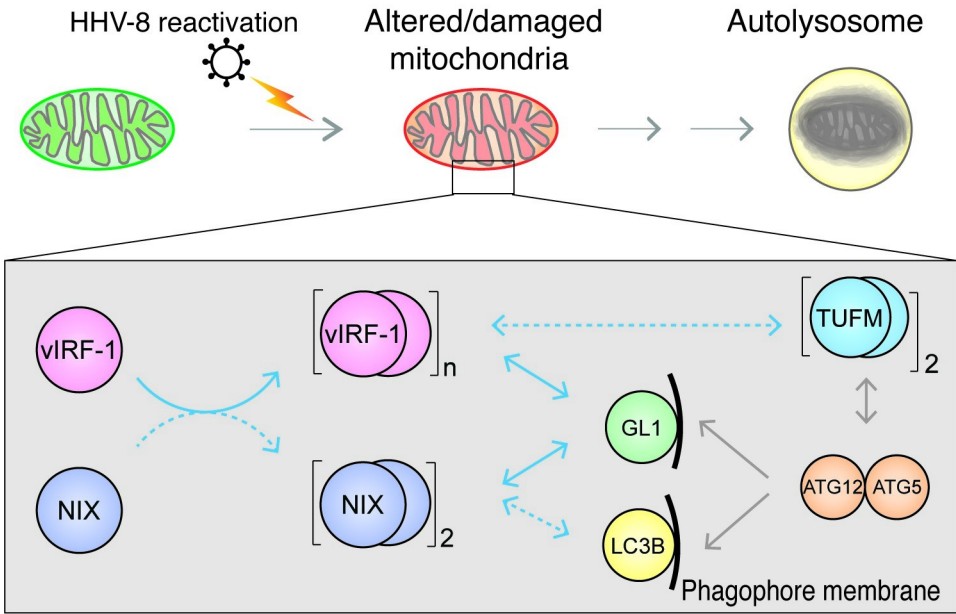

**Fig 10. Proposed models of vIRF-1-mediated mitophagy.** vIRF-1 is inducibly expressed during lytic replication and targeted to altered or damaged mitochondria, where it plays roles in mitophagy initiation and progression, likely via activation of NIX and TUFM-mediated mitophagy and recruitment of the autophagosomal membrane-bound GABARAPL1 to the mitochondria. Solid and dashed blue arrows indicate the pathways identified by the current study and previously published findings [16,17,18]. Solid and dashed gray arrows indicate the mechanisms identified by others [29,32,47] and unknown, respectively. GL1, GABARAPL1; 'n', multimer; '2', dimer.

in lytically infected cells. Furthermore, we found that overexpression of the cellular mitophagy receptor NIX enhances the interaction of vIRF-1 with GABARAPL1, potentially by promoting vIRF-1 dimerization and then stabilization of oligomerized vIRF-1. Importantly, we identified a short novel amino acid sequence motif of vIRF-1 required for GABARAPL1 binding. However, overexpression of vIRF-1 alone does not induce mitophagy flux and mitochondrial clearance [17]. These findings suggest that vIRF-1 plays a role as a mitophagy mediator rather than a mitophagy receptor.

Besides vIRF-1, other viral proteins that can interact with ATG8 proteins have been reported [8,38]. For example, the matrix protein M2 of influenza A virus has been verified experimentally to contain a conserved LIR motif, which is required for binding to and redistributing LC3 to the plasma membrane, thereby subverting antiviral autophagy (virophagy) [39]. The lytic gene product BALF1 of Epstein-Barr virus is known to bind directly via its LIR motif to LC3 and to activate autophagy [40]. The two nonstructural proteins, NSP15 and NSP12, of SARS-CoV-2, are predicted to contain putative LIR motifs and are co-localized with LC3B or GABARAP in transfected cells, although their direct interactions remain to be demonstrated [38]. In addition to the LIR-dependent interactions, the viral infectivity factor (Vif) of HIV-1 binds directly to LC3, preferentially LC3-I over the lipidated form LC3-II, via a region (residues 144–159) inside the SOCS box-like domain, which lacks a conserved LIR [41]. Interestingly, the matrix M protein of human parainfluenza virus type 3 (HPIV3) is known to interact with LC3 and TUFM, and the interactions are required for HPIV3 infection-induced mitophagy [32]. The M protein has no conserved LIR; a lysine at position 295 in the C-terminus is critical for LC3 interaction. Overexpression of the M protein activates mitophagy flux and induces mitochondrial clearance. These results suggest that HPIV3 M also plays a role as a viral mitophagy receptor that connects TUFM (mitochondria) to LC3-containing

phagophores [32]. In contrast to HPIV3 M, however, overexpression of vIRF-1 alone does not induce mitophagy. In addition, vIRF-1 is unique in that it can be targeted directly to the OMM via direct interactions with lipids, such as cardiolipin and cholesterol [16], interact preferentially with GABARAPL1, and be activated by NIX. Thus, each likely represents a different class of viral mitophagy mediator. Nonetheless, it is noteworthy that TUFM is critically involved in mitophagy pathways via these two viral mitophagy mediators.

vIRF-1 appears to bind selectively to the lipidated forms of GABARAPL1 in cells (Fig 4D). However, our *in vitro* binding assays showed that the non-lipidated form of GABARAPL1 expressed as GST-fusion protein in bacteria could bind to vIRF-1 (Fig 1B). This discrepancy might be due to a cellular factor or condition (different localization) that inhibits the interaction between ATG8 proteins and vIRF-1 within the cell. As vIRF-1 binds selectively to GABARAPL1 in virus-infected cells, we focused on GABARAPL1 for further structural and functional analyses and identified three critical amino acids, E17, K20, and K38, required for vIRF-1 interaction. E17 is conserved among ATG8 proteins except for LC3B, K20 in the GABARAP subfamily, and K38 in GABARAP and GABARAPL1. From the sequence analysis, it is expected that vIRF-1 could also interact with GABARAP, as demonstrated experimentally in Fig 1B and 1C. However, the interaction of vIRF-1 with GABARAP was weaker than with GABARAPL1 in reactivated iBCBL-1 cells (Fig 1C). This different result might be dependent on cell types. Thus, further study is warranted to compare the ability of vIRF-1 to bind to the GABARAP proteins in other HHV-8-infected cells, including iSLK and endothelial cells.

What are the functional implications of the interactions of vIRF-1 with GABARAP-subfamily proteins? According to the literature [7,42], the two ATG8 subfamilies may play different roles in autophagosome biogenesis; the GABARAP subfamily may act at the initiation or a later stage in the process whereas the LC3 subfamily is essential for phagophore elongation. GABARAP and GABARAPL1 can, more efficiently than LC3B, bind to and recruit the Unc-51-like kinase (ULK) complex, consisting of ULK1, ULK2, ATG13, FIP200, and ATG101, and the autophagy class III phosphatidylinositol 3-kinase complex I (PI3K-C1), consisting of VPS34, VPS15, BECN 1, and ATG14, to the site of phagophore nucleation [22,43]. Moreover, GABARAP proteins can bind to TBC1D15, a mitochondrial Rab GTPase-activating protein, and mediate autophagic encapsulation of mitochondria by inhibiting the small GTPase Rab7 and promoting the closure of autophagosomal membranes [44]. Therefore, further study is warranted to examine whether vIRF-1 plays a role in phagophore nucleation or closure of autophagosomal membranes by promoting the recruitment of the GABARAP-containing autophagy machinery complexes to mitochondria for mitophagy activation.

How does the mutation in the NLS of vIRF-1 affect the formation of aggregated vIRF-1? Immunoblotting analysis shows that aggregated forms of the vIRF-1 NLS$^X$ variant could be highly detected in the mitochondria isolated from NIX-overexpressing cells compared to those of native vIRF-1. This might be simply due to a higher concentration of the NLS$^X$ variant at the cytoplasm than native vIRF-1. In supporting this notion, aggregated forms of native vIRF-1 in virus-infected cells were readily detected when highly expressed 2 and 3 days after reactivation (Fig 8A). Nonetheless, we don't exclude the possibility that the mutation at the NLS sequence (RGRRR) of vIRF-1 makes it prone to aggregate in the presence of NIX.

## Materials and methods

### Cell culture

293T, HeLa.Kyoto and iSLK cells were cultured in DMEM media supplemented with 10% fetal bovine serum and 1% antibiotics of penicillin and streptomycin in a humidified incubator at 5% $CO_2$ at 37°C. TRExBCBL-1-RTA (here termed iBCBL-1) cells were cultured in RPMI-1640

medium supplemented with 15% fetal bovine serum, 1% antibiotics, and 0.1% plasmocin prophylactic (InvivoGen, ant-mpp). HeLa.Kyoto cells were kindly provided by Ron R. Kopito. iBCBL-1 and iSLK cells were kindly provided by Jae U. Jung.

## Transfection and transduction

Transient transfection with plasmids was performed using GenJet DNA transfection reagent (SignaGen Laboratories, SL100489) according to the manufacturer's instruction. Lentiviral transduction was performed as before for stable expression of the mitophagy reporter mito-mCE, shRNAs, and sgRNAs [45]. In brief, to produce infectious lentiviruses, 293T cells were co-transfected with a lentiviral transfer vector together with the packaging plasmid psPAX2 (a gift from Didier Trono, Addgene plasmid #12260) and the vesicular stomatitis virus G protein expression plasmid pVGV-G at a ratio of 5:4:1 for 2 days. Transduction units (TUs) of 60x concentrated lentiviruses were determined in 293T cells in the presence of appropriate antibiotics to select transduced cells. Target cells were transduced with lentiviruses at a single dose of 10 TUs in the presence of 10 μg/ml polybrene for 6 h. Stably transduced cells were selected for 2–3 weeks in the presence of appropriate antibiotics.

## DNA manipulation

All polymerase chain reaction amplification and site-directed mutagenesis, including point and deletion mutations, were performed using SuperFi DNA polymerase (Thermo Fisher Scientific). Subcloning of open reading frames and their derivatives into expression plasmids, including pICE (a gift from Steve Jackson, Addgene plasmid #46960), pGEX-4T-1 (GE Healthcare Life Sciences), pLenti.puro (a gift from Melina Fan, Addgene plasmid #74218), and Nano-BiT system vectors (Promega), was performed using appropriate restriction enzyme sites. In addition, the LentiCRISPR v2 vector (a gift from Feng Zhang, Addgene plasmid #52961) was used for the lentiviral transduction of GABARAPL1 sgRNA into HeLa.Kyoto, iSLK.BAC16, iBCBL-1 cells, and NIX sgRNA into iBCBL-1 cells.

## Antibodies

T7 tag (69-048-3MI) antibody was purchased from Novagen. GST (sc-138), LDH (sc-33781), FLOT1 (sc-74566), NIX (sc-166314), TUFM (sc-393924), and TOM20 (sc-17764) antibodies were from Santa Cruz Biotechnology. Flag (M2, F3165) and HA (3F10, 11867423001) antibodies were from Sigma-Aldrich. GABARAPL1 (66458–1), β-Actin (60008–1), Lamin B1 (12987–1), and MT-ND1 (19703–1) antibodies were from Proteintech. V5 tag (R960-25) antibody was from Thermo Fisher Scientific. LC3B (NB100-2220) antibody was from Novus Biologicals. LC3A (4599), LC3C (14736), GABARAPL1 (26632), GABARAPL2 (14256S), VDAC (4661), Flag tag (14793), and V5 tag (13202) antibodies were from Cell Signaling Technology. P62/SQSTM1 (PM045) and GABARAP (PM037) antibodies were from MBL International. MT-CO2 antibody (ab79393) was from Abcam. LAMP1 antibody (11215-R107) was from Sino Biological. VDAC (600-101-HB2) antibody was from Rockland. Anti-RTA and rabbit anti-vIRF-1 were a gift from Gary Hayward, and rat anti-vIRF-1 (residues 1–22) was custom-generated from Thermo Fisher.

## Immunological assays

For the preparation of whole cell extracts, cells were resuspended in RIPA buffer (50 mM Tris [pH 7.4], 150 mM NaCl, 1% Igepal CA-630, and 0.25% deoxycholate) containing a protease inhibitor cocktail and protein phosphatase inhibitors (10 mM sodium fluoride and 5 mM

sodium orthovanadate) and sonicated using Bioruptor (Diagenode) for 5 min in ice water at a high-power setting (320 W). For immunoblotting, cell lysates and protein samples were separated by SDS-PAGE, transferred to nitrocellulose or polyvinylidene difluoride membranes, and immunoblotted with appropriate primary antibodies diluted in SuperBlock (phosphate-buffered saline (PBS)) blocking buffer (Thermo Fisher Scientific). Following incubation with horse radish peroxidase-labeled appropriate secondary antibody, immunoreactive bands were visualized by enhanced chemiluminescent (ECL) reagents, Clarity (Bio-Rad) and SuperSignal West Femto (Thermo Fisher Scientific) on an ECL film. ImageJ software (National Institute of Health) was used to quantify the intensities of bands from immunoblots. For Flag-immuno-precipitation, whole cell extracts were incubated with affinity gel (anti-DYKDDDDK tag (L5) gel, BioLegend, 651502) at 4˚C overnight. Immunoprecipitants were washed with RIPA buffer, and bound proteins were eluted by boiling in 1× SDS sample buffer. A Clean-Blot IP detection reagent (Thermo Fisher Scientific) was used to avoid the detection of IgG in IP assays. For indirect immunofluorescence assay (IFA), HeLa.Kyoto cells grown on a coverslip were transfected and permeabilized with 25 µg/ml of saponin in PBS containing 100 mM potassium chloride before fixation in Image-iT fixative solution (Thermo Fisher Scientific) or permeabilized in 0.5% Triton X-100 in PBS after fixation. As above, reactivated iBCBL-1 cells were attached to poly-L-lysine-coated coverslips and permeabilized after fixation. Following incubation with SuperBlock™-PBS blocking buffer for 1 h at room temperature, coverslips were incubated with primary antibodies, washed with PBS, and then incubated with appropriate fluorescent dye-conjugated secondary antibodies. Coverslips were mounted on a glass slide in ProLong Gold Antifade Mountant containing DAPI (Thermo Fisher Scientific). Cells were imaged by Zen software on a Zeiss confocal laser scanning microscope 700 with an oil immersion x40 or x63 objective.

## Manipulation of HHV-8 bacmid BAC16

The BAC16 genome was edited using a two-step seamless Red recombination in the bacteria strain GS1783 (a gift from Greg Smith) as previously performed [17]. The recombinant BAC DNAs were purified using the NucleoBond BAC kit (Clontech). The recombination area was amplified by PCR, and the deletion mutation of the vIRF-1 GIR was verified by DNA sequencing of the PCR amplicon. Gross genome integrity was examined on agarose gel electrophoresis after BspHI digestion of the BAC16 DNAs. For the production of BAC16-infected iSLK cell lines, co-culture infection was used. Briefly, the 293T cells stably transfected with the recombinant BAC16 DNAs using Lipofectamine 2000 (Invitrogen) were incubated together with naïve iSLK cells in a 1:1 ratio in the presence of 30 nM phorbol ester (TPA) and 300 nM sodium butyrate for 5 days. Then, infected iSLK cell lines were selected by culturing in the presence of 1 mg/ml hygromycin B, 1 µg/ml puromycin, and 250 µg/ml G418.

## Cell fractionation

To fractionate cells into the nuclear and cytoplasmic fractions, we homogenized cells in buffer B (0.25M sucrose, 1 mM EDTA, 20 mM HEPES-NaOH [pH 7.4]) with 50 strokes of a Dounce glass homogenizer and centrifuged at 1,000 x g for 10 min. The supernatant was added to the equal volume of 2x RIPA buffer for immunoprecipitation and used as the cytoplasmic fraction. The pellet was washed in buffer B, resuspended in 1x RIPA buffer, sonicated, and used as the nuclear fraction. For mitochondria isolation, the homogenate supernatant was further centrifuged at 13,000 x g for 10 min. Mitochondria were enriched from the pellet using iodixanol as previously performed [17].

## Mitophagy flux assays

HeLa.Kyoto cells stably expressing the mitophagy fluorescence reporter mito-mCE were imaged using the ZOE fluorescent cell imager (Bio-Rad, Hercules, CA) and Zeiss confocal laser scanning microscope 700. ImageJ (Fiji) was used for all image preparation and analysis. To determine a ratio of red to green fluorescence, each cell in a merged RGB image was defined using the regions of interest tool. The intensities of red and green fluorescence were measured using the color histogram tool.

## GST pull-down assays

Bacterially expressed recombinant glutathione-S-transferase (GST) and GST-fusion proteins were purified by standard methods. First, 1 μg GST and GST-fusion proteins were incubated with 20 μl (1:1 slurry) of glutathione sepharose-4B beads in binding buffer (PBS plus 1% Triton X-100) for 1 h at room temperature. After washing with binding buffer four times, the protein-bead complexes were incubated with purified recombinant proteins or cell lysates as described previously [23], then washed in binding buffer four times, separated by SDS-PAGE, and subjected to immunoblotting.

## Protein fragment complementation NanoBiT assay

NanoBiT assays were performed as previously described [17,45]. In brief, twenty-four hours after transfection of 293T cells in a 6-well plate with the indicated genes in the NanoBiT binary vectors, cells were washed in PBS and resuspended in 1 ml of Opti-MEM I medium (Thermo Fisher Scientific), and 100 μl of the cell suspension was transferred to a 96-well opaque plate in triplicate. Furimazine (Promega), a NanoLuc-luciferase substrate, was diluted in PBS at 1:100, and 25 μl of the diluted substrate was added to each well. After 5 min of incubation, the luminescence was measured by a GloMax 96 microplate luminometer (Promega, Madison, WI).

## Real-time-quantitative PCR (RT-qPCR) analysis

Total RNAs were isolated using the Quick-RNA Miniprep Plus kit (Zymo Research). First-strand cDNA was synthesized from 1 μg of total RNA using SuperScript IV reverse transcriptase (Thermo Fisher Scientific) with random hexamers. Next, RT-qPCR was performed using QuantStudio 3 (Thermo Fisher Scientific) with the FastStart SYBR green/ROX master mix (Sigma-Aldrich). Primers are listed in S1 Table. Reactions were performed in a total volume of 10 μl and contained 50 ng of reverse-transcribed RNA (based on the initial RNA concentration) and gene-specific primers. PCR conditions included an initial incubation step of 2 min at 50˚C and an enzyme heat activation step of 10 min at 95˚C, followed by 40 cycles of 15 s at 95˚C for denaturing and 1 min at 60˚C for annealing and extension.

## HHV-8 replication assays

For the determination of encapsidated HHV-8 genome copy number, viral DNA was purified using Quick-DNA Viral Kit (Zymo Research) following pretreatment of the culture supernatants containing HHV-8 virions with DNase I (New England BioLabs) at 37˚C overnight. RT-qPCR was performed as described above with LANA primers. BAC16 DNA was used as a standard for the calibration curve [17].

## Reagents

Doxycycline hyclate (D5207), cycloheximide (C7698), saponin (S4521), and leupeptin (L5793) were purchased from Sigma-Aldrich.

## Statistical analyses

The one-way ANOVA test was used to assess the statistical significance of differences between groups, and Student's t-test was used for post hoc pairwise comparisons. The statistical significance was considered significant when the *p-value* was less than 0.05 ($p < 0.05$).

## Supporting information

**S1 Fig. Analysis of the protein and mRNA expression of ATG8 proteins in latent and lytic iBCBL-1 cells. (A)** Immunoblots of extracts derived from iBCBL-1 cells treated with Dox for 0 to 3 days. Relative expression of each ATG8 protein was determined by dividing by the band intensity at 0 day and depicted in the bar graphs (right). **(B)** RT-qPCR analysis of the mRNA expression of ATG8 genes. iBCBL-1 cells were treated with or without Dox for 2 days. *, $p < 0.05$; ns, not significant.
(TIF)

**S2 Fig. Sequence analyses of human ATG8 proteins using the web tool Clustal Omega. (A)** Sequence alignment of ATG8 proteins. Compared to the other ATG8 members, seven blocks of GABARAPL1 containing relatively poorly conserved 6 residues were selected and replaced each corresponding block of LC3B with the counterparts of GABARAPL1. **(B)** Phylogenetic tree of ATG8 proteins.
(TIF)

**S3 Fig. Confocal images of the mitophagy reporter HeLa.Kyoto mito-mCE cell line.** The cell lines were transfected with the indicated plasmids expressing V5-GABARAPL1, vIRF-1-Flag, and HA-NIX for 24 h in the presence of 40 μM leupeptin. The enlarged images of the insets are shown in Fig 5B. Scale bar, 10 μm.
(TIF)

**S4 Fig. Comparative analysis of the expression levels of endogenous NIX protein in HeLa.Kyoto and iBCBL-1 cells.** Immunoblots of extracts derived from HeLa.Kyoto transfected with empty vector (EV) or HA-NIX and iBCBL-1 cells treated with or without Dox for 2 days. Endogenous NIX protein in HeLa.Kyoto cells was hardly detected compared to those in NIX-transfected HeLa.Kyoto and iBCBL-1 cells. NIX antibody (D4R4B) was purchased from Cell Signaling Technology.
(TIF)

**S5 Fig. GABARAPL1 plays an important role in NIX/vIRF-1-mediated mitophagy via the VIR, HP, and G116 regions.** Mitophagy assays in control and *GABARAPL1* KO HeLa.Kyoto cells co-transfected with V5-NIX, vIRF-1-Flag, or HA-GABARAPL1 WT and variants (VIR^X, HP^X, and G116A) along with the reporter mito-mCE. More than 50 cells per sample were counted. The one-way ANOVA test was used to assess the statistical significance of differences between groups, and the t-test was used for post hoc pairwise comparisons. *, $p < 0.05$; ** $p < 0.01$; and ***, $p < 0.001$.
(TIF)

**S1 Table. Sequences of oligonucleotides used in this study.**
(XLSX)

## Acknowledgments

We appreciate Drs. John Nicholas and Edward W. Harhaj for proofreading the paper.

## Author Contributions

**Formal analysis:** Mai Tram Vo, Young Bong Choi.

**Funding acquisition:** Young Bong Choi.

**Investigation:** Mai Tram Vo, Chang-Yong Choi, Young Bong Choi.

**Methodology:** Mai Tram Vo, Chang-Yong Choi, Young Bong Choi.

**Project administration:** Young Bong Choi.

**Supervision:** Young Bong Choi.

**Validation:** Young Bong Choi.

**Writing – original draft:** Mai Tram Vo.

**Writing – review & editing:** Young Bong Choi.

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
