## [Decision Letter · Decision Letter 0]

25 Mar 2023

Dear Ph.D. Choi,

Thank you very much for submitting your manuscript "The mitophagy receptor NIX induces vIRF-1 oligomerization and interaction with GABARAPL1 for the promotion of HHV-8 reactivation-induced mitophagy" for consideration at PLOS Pathogens. As with all papers reviewed by the journal, your manuscript was reviewed by members of the editorial board and by several independent reviewers. In light of the reviews (below this email), we would like to invite the resubmission of a significantly-revised version that takes into account the reviewers' comments.

Overall, reviewers were enthusiastic about the manuscript and recognize the importance of mechanistic studies of vIRF-1 function to mediate mitophagy that promotes KSHV production, here the interaction with the autophagy-related gene 8 protein GABARAPL1. However, each noted numerous concerns that need to be carefully considered and addressed before further consideration for publication. Reviewers note that data is often inconsistent and descriptions of the results are at odds with the data shown; key statistical analyses were missing or considered inappropriate. In particular, there were concerns about the lack of IP experiments in the context of more biologically relevant conditions, comparisons made with variable expression levels, and the lack of validation with recombinant viruses. The absence of data demonstrating a significant impact of these protein interactions in promoting mitophagy in the context of virus reactivation was a shared concern.

We cannot make any decision about publication until we have seen the revised manuscript and your response to the reviewers' comments. Your revised manuscript is also likely to be sent to reviewers for further evaluation.

Sincerely,

Laurie T Krug, PhD

Academic Editor

PLOS Pathogens

Alison McBride

Section Editor

PLOS Pathogens

Kasturi Haldar

Editor-in-Chief

PLOS Pathogens

orcid.org/0000-0001-5065-158X

Michael Malim

Editor-in-Chief

PLOS Pathogens

orcid.org/0000-0002-7699-2064

Overall, reviewers were enthusiastic about the manuscript and recognize the importance of mechanistic studies of vIRF-1 function to mediate mitophagy that promotes KSHV production, here the interaction with the autophagy-related gene 8 protein GABARAPL1. However, each noted numerous concerns that need to be carefully considered and addressed before further consideration for publication. Reviewers note that data is often inconsistent and descriptions of the results are at odds with the data shown; key statistical analyses were missing or considered inappropriate. In particular, there were concerns about the lack of IP experiments in the context of more biologically relevant conditions, comparisons made with variable expression levels, and the lack of validation with recombinant viruses. The absence of data demonstrating a significant impact of these protein interactions in promoting mitophagy in the context of virus reactivation was a shared concern.

Reviewer's Responses to Questions

**Part I - Summary**

Reviewer #1: In their previous work, the Choi group showed that the KSHV viral interferon regulatory factor 1 (vIRF1) binds to the autophagy receptor NIX on the outer membrane of the mitochondria to induce mitophagy and support lytic replication. In this manuscript, they explore the mechanism behind vIRF1-NIX-mediated mitophagy. They show that vIRF1 binds to the autophagy-related gene 8 (ATG8) family of proteins (central modifiers for autophagy) using recombinant proteins, with a strong preference for the GABARAPL1 in KSHV-infected cells. Through biochemical assays, they systematically mapped this interaction to 10 residues on vIRF1 and 3 residues on GABARAPL1. This vIRF1-GABARAPL1 interaction increased mitophagy in HeLa cells and further enhanced in the presence of NIX. Furthermore, NIX induces the oligomerization and stabilization of vIRF1 on the mitochondrial surface. Together, they propose a model where, during lytic reactivation, NIX is activated and oligomerizes vIRF1 on the mitochondrial surface. Oligomerized NIX and vIRF1 bind to ATG8 proteins (e.g. GABARAPL1) for mitophagy. The manuscript is well-written, and experiments are logical and well-controlled.

Reviewer #2: The manuscript by Choi and colleagues investigates the mechanism of vIRF1-induced mitophagy during KSHV lytic infection. The authors describe an interaction between vIRF1 and the autophagy related protein, GABARAPL1, and identify key residues involved in the interaction. The mitophagy receptor NIX was found to facilitate the interaction. a vIRF1 variant defective in GABARAPL1 binding had reduced ability to promote vIRF1-NIX-induced mitophagy. Lastly, a NIX CRISPR KO was found to exhibit reduced KSHV reactivation. Overall, the study provides an in depth characterization between the interaction between vIRF1 and GABARAPL1, with interesting hints to biological significance.

Reviewer #3: In this study, Vo et al. analyzes the molecular mechanism of how vIRF1 promotes mitophagy during the lytic cycle of KSHV, which is important for efficient virus production. This study is based on the previous findings of the Choi lab. Here, they demonstrate that vIRF1 interacts with the autophagy protein GABARAPL1, which interaction in association with the mitophagy receptor NIX is required for vIRF1-mediated mitophagy. Overall, this is very interesting study, which provides new information about the regulation of mitophagy by a KSHV protein. While most of the experiments were well executed, in some cases I found that the interpretation of some results can be improved and adding a few more data to some experiments would be helpful for better understanding the results.

Comments:

Line 124-125. Instead of “lytically HHV-8 infected”, I suggest using TRExBCBL-1-RTA cells undergoing lytic reactivation or during lytic reactivation using XY cells…Because lytic reactivation was induced and not lytic KSHV infection of the cells.

GSTPD in Fig 1B shows direct interaction of vIRF1 with LC3C, GABARAP, GABARAPL1 and GABARAPL2 but only GABARAPL1 interacts with vIRF1 in cells. How can they explain it? Maybe instead of using whole cell lysate, cytoplasmic extract should be used for IP.

Fig 1D. Quality of IFA is low and lytic reactivation seems to be inefficient. 2 dpi should show much more vIRF1 expression, it is barely detectable in cells, and it is exclusively cytoplasmic. Should there not be nuclear vIRF1staining as well? Like in their previous Nature Communication paper.

Fig 1F. Does the vIRF1 mutant interact with GABARAPL1 in cells? IP using cell lysate seems to be more stringent assay to test interaction than GSTPD based on Fig 1B and C. Either using transfected 293T cells or BCBL1 expressing vIRF1 WT or mutant (by lenti).

Line 140-141. Lack of interaction is a confirmation of the predicated motif not being involved in ATG8 protein interaction? If G and R are not tolerated in LIR motif, how DWGRL was predicted as LIR motif in the first place?

Fig 2B. Why DBD-IAD does not interact with GABARAPL1 if DBD of vIRF1 is required for their interaction? What is the band in the first lane of the input?

Line 171. Subtitle is missing for the description of Fig 3, which shows the mapping of interaction domain in GABARAPL1.

Fig 5F. This is an important experiment, but its interpretation is confusing. It was described in the Results section only briefly. I note that HA-GABARAPL1 expression in the complemented KO cells is much more than the expression of endogenous GABARAPL1 in WT HeLa cells. Does this affect mitophagy and interpretation of the results? The authors claim that mitophagy was significantly inhibited in the KO cell line but was rescued by WT GABARAPL1 expression. I do not see it if I compare lane 5 with lane 1 and lane 6 with lane 5. There are some small differences, but I do not know if they are significant. I do not understand why the investigators show the significant tests for those they show. For example, comparing lane 11 with lane 2 basically show the effect of overexpressed HA-GABARAPL1 on mitophagy, which is the same as shown in lane 5 in Fig 5C. To my opinion the samples that would be really important to compare with each other are in lane 1 (WT cells, mock), 4 (WT cells with vIRF1/NIX), 5 (KO cells, mock), 20 (KO cells with vIRF1/NIX), and 21 (KO cells complemented with GABARAPL1 and vIRF1/NIX expressed). However, to make unambiguous conclusion, it would be important to have the same expression level of GABARAPL1 across different samples/cell lines. Alternatively, HA-GABAPARL1 should also be overexpressed in WT HeLa cells and use these cells for comparison.

Does Fig 5C show one single blot (24-25-well protein gel?) or 3 blots together? If several blots were put together, I suggest using a line between the blots.

Fig 6C. vIRF1 runs at 75 kDa? It should be around 50 kDa as shown on other blots.

Fig 6D. If the NLS mutant of vIRF1 is not co-localized with mitochondria, how this vIRF1 mutants can induce more mitophagy? Does the NLS mutant interact (more/stronger) with GABARAPL1 and NIX?

Fig 7B. vIRF1 protein seems to be extremely stable that even 8 hours of CHX treatment did not change its protein level. If this is indeed the case, lane 339 “modified and aggregated vIRF1 exhibit a high protein turnover” is wrong. It should read low protein turnover.

How does the mutation in NLS or GABARAPL1 binding domain of vIRF1 affect the formation of aggregated vIRF1? How is this protein aggregation that does not seems to be associated with mitochondria connected to mitophagy?

**Part II – Major Issues: Key Experiments Required for Acceptance**

Reviewer #1: However, a couple of major points are unclear:

1. Is GABARAPL1 also required for vIRF1 oligomerization similar to NIX in the context of infection/iBCBL-1? Do you also see oligomerization of GABARAPL1?

2. Although the authors have shown that GABARAPL1 significantly increases mitophagy induced by vIRF1 and NIX (Fig. 5) in HeLa cells, this needs to be demonstrated in the context of infection/iBCBL-1. Is GABARAPL1 and its interaction with vIRF1 important for lytic reactivation? More specifically, is the interaction between vIRF1 and GABARAPL1 crucial for mitophagy during reactivation? Does this interaction have an impact on viral progeny? It might be useful to express the GABARAPL1 mutants in iBCBL-1 (in the context of endogenous GABARAPL1 KO) to see if they can rescue the phenotypes?

Reviewer #2: As presented, interpretation of the data presents challenges due to a lack of consistency. For example, interactions between LC3B and vIRF1 are observable in roughly half of the figures. While this may be due to the intensity of the blots, this needs to be corrected to present a consistent line of data for interpretation.

The experimental setup of Figure 3 is unclear and would benefit from additional text. For example, why was LC3B chosen for this pairwise alignment? How well are these two proteins conserved? LC3A also does not interact well with vIRF1 in vitro--would doing this analysis with LC3A produce a similar set of regions?

Further investigation of how a lack of vIRF1 binding to GABARAPL1 would also be useful. For instance, through the construction of the vIRF1 227-236 deletion BAC and its characterization. Or complementing the vIRF1 KO with the mutant.

While the vIRF1 aggregation is interesting, its biological importance is unclear. Comments describing it as an essential step in mitophagy activation are unwarranted.

The phenotype for NIX KO is minor—along this line, mitolysosomes only appear to form in a few cells—13 out of 257. Interpretation of this is confounded by the fact that lytic reactivation of iBCBL1 cells is not efficient, traditionally reported to occur for roughly half of cells in the population. Are the mitolysomes reported in Fig S4 in lytic cells? Staining for vIRF1 or another lytic marker would be sufficient.

Reviewer #3: (No Response)

**Part III – Minor Issues: Editorial and Data Presentation Modifications**

Reviewer #1: Minor points that can be addressed to improve the manuscript:

1. Lines 160-164: A short explanation of the NanoBiT assay would be helpful in the main text.

2. Fig. 5D needs to be explained more in the main text. It was barely mentioned in lines 249-253.

3. Fig. 7C: IC50 is not an appropriate calculation to measure protein half-life (this is not used for time measurements as a variable). A decay/half-life function should be used.

4. For majority of the statistical tests (Figs. 2G, 5A, 5F, 5I, 7A), ANOVA should be used and not t test since there are >2 groups. Appropriate posthoc tests can be done to calculate pairwise adjusted p values.

5. The authors may want to consider placing Fig. S4 as a main figure.

Reviewer #2: (No Response)

Reviewer #3: (No Response)

PLOS authors have the option to publish the peer review history of their article (what does this mean?). If published, this will include your full peer review and any attached files.

Reviewer #1: No

Reviewer #2: No

Reviewer #3: No

Figure Files:

Data Requirements:

Please note that, as a condition of publication, PLOS' data policy requires that you make available all data used to draw the conclusions outlined in your manuscript. Data must be deposited in an appropriate repository, included within the body of the manuscript, or uploaded as supporting information. This includes all numerical values that were used to generate graphs, histograms etc.. For an example see here on PLOS Biology: http://www.plosbiology.org/article/info:doi%2F10.1371%2Fjournal.pbio.1001908#s5.
---

## [Decision Letter · Decision Letter 1]

7 Jul 2023

Dear Ph.D. Choi,

We are pleased to inform you that your manuscript 'The mitophagy receptor NIX induces vIRF-1 oligomerization and interaction with GABARAPL1 for the promotion of HHV-8 reactivation-induced mitophagy' has been provisionally accepted for publication in PLOS Pathogens.

Best regards,

Laurie T Krug, PhD

Academic Editor

PLOS Pathogens

Alison McBride

Section Editor

PLOS Pathogens

Kasturi Haldar

Editor-in-Chief

PLOS Pathogens

orcid.org/0000-0001-5065-158X

Michael Malim

Editor-in-Chief

PLOS Pathogens

orcid.org/0000-0002-7699-2064

Reviewer Comments (if any, and for reference):

Reviewer's Responses to Questions

**Part I - Summary**

Reviewer #1: The authors have adequately addressed my concerns and comments. The new data in Fig. 5 (GABARAPL1 KO in iSLKBAC16 as well as using a vIRF1 dGIR BAC16 mutant) provided more convincing data about the relevance of the interactions.

Reviewer #2: The authors have been highly responsive to all reviews and the manuscript is greatly improved. My concerns have all been addressed.

Reviewer #3: The authors have adequately addresses my questions.

**Part II – Major Issues: Key Experiments Required for Acceptance**

Reviewer #1: (No Response)

Reviewer #2: (No Response)

Reviewer #3: (No Response)

**Part III – Minor Issues: Editorial and Data Presentation Modifications**

Reviewer #1: (No Response)

Reviewer #2: (No Response)

Reviewer #3: (No Response)

PLOS authors have the option to publish the peer review history of their article (what does this mean?). If published, this will include your full peer review and any attached files.

Reviewer #1: No

Reviewer #2: No

Reviewer #3: No

---

## [Editor Report · Acceptance letter]

12 Jul 2023

Dear Ph.D. Choi,

We are delighted to inform you that your manuscript, "The mitophagy receptor NIX induces vIRF-1 oligomerization and interaction with GABARAPL1 for the promotion of HHV-8 reactivation-induced mitophagy," has been formally accepted for publication in PLOS Pathogens.

Best regards,

Kasturi Haldar

Editor-in-Chief

PLOS Pathogens

orcid.org/0000-0001-5065-158X

Michael Malim

Editor-in-Chief

PLOS Pathogens

orcid.org/0000-0002-7699-2064